



# Under-resolved gradients: slow wake recovery and fast turbulence decay with mesoscale Wind Farm Parameterizations

William C. Radünz[1], Jens H. Kasper[2], Richard J. A. M. Stevens[2], and Julie K. Lundquist[1,3]

[1]Johns Hopkins University, Baltimore, MD, USA
[2]Physics of Fluids Group, Max Planck Center for Complex Fluid Dynamics, J. M. Burgers Center for Fluid Dynamics, University of Twente, Enschede, the Netherlands
[3]National Renewable Energy Laboratory, Boulder, CO, USA

**Correspondence:** William C. Radünz (wcorrea1@jh.edu)

**Abstract.** Numerical Weather Prediction (NWP) and climate models equipped with Wind Farm Parameterizations (WFPs)
can simulate cluster wake effects affecting downstream wind farms in both onshore and offshore environments. This study
evaluates the strengths and limitations of the NWP-WFP approach using the Weather Research and Forecasting (WRF) model
with the Fitch WFP, benchmarked against large-eddy simulations (LES) of an idealized offshore wind farm under neutral
atmospheric stability. Wake recovery is underestimated in NWP-WFP simulations because of two interconnected issues. First,
the spatial gradients in the wind velocity field within the near-farm wake are necessarily under-resolved at mesoscale grid
resolutions compared to the LES. Second, the faster decay of farm-added turbulence kinetic energy (TKE) in the mesoscale
simulations is likely not due to excessive dissipation, but rather due to the underestimation of spatial gradients needed to sustain
elevated TKE levels via shear production. A key insight is that the slow recovery in the near-farm wake, although confined
to a short downstream distance, has lasting consequences for the far wake region. This fact underscores the need to address
under-resolved spatial gradients to improve wake recovery and reduce far wake biases in NWP-WFP simulations.
*Copyright statement.* This work was authored in part by the National Renewable Energy Laboratory, operated by Alliance for Sustainable
Energy, LLC, for the U.S. Department of Energy (DOE) under contract no. DE-AC36-08GO28308. Funding was provided by the U.S. De-
partment of Energy Office of Energy Efficiency and Renewable Energy Wind Energy Technologies Office. This work was partially supported
by the U.S. Department of Energy, Office of Science Energy Earthshot Initiative, as part of the Addressing Challenges in Energy—Floating
Wind in a Changing Climate (ACE-FWICC) Energy Earthshot Research Center. The views expressed in the article do not necessarily repre-
sent the views of the DOE or the U.S. Government. The U.S. Government retains and the publisher, by accepting the article for publication,
acknowledges that the U.S. Government retains a nonexclusive, paid-up, irrevocable, worldwide license to publish or reproduce the published
form of this work, or allow others to do so, for U.S. Government purposes.

## 1 Introduction

The sheer scale of offshore wind farms is attracting increasing attention from the scientific community (Barthelmie et al.,
2007, 2009; Platis et al., 2018; Pryor et al., 2020; Cañadillas et al., 2020; Rosencrans et al., 2024; Ouro et al., 2025) largely





due to cluster wake effects and associated power losses. The combination of massive wind farms (∼1 GW), large turbines
(10–20+ MW), and the relatively low turbulence offshore (Bodini et al., 2019) leads to strong, persistent wakes that can extend
tens of kilometers downstream of the wind farms (Platis et al., 2018). As a result, inter-farm wake effects pose challenges not
only to operating wind farms but also to those in planning or development stages, thus mirroring concerns already observed in
onshore settings (Lundquist et al., 2019). Understanding and accurately predicting the wakes that arise from atmosphere–wind
farm interactions is therefore of scientific and economic relevance (Veers et al., 2019, 2022). In this context, Numerical Weather
Prediction (NWP) models equipped with Wind Farm Parameterizations (WFPs) are proving to be a valuable tool (Fischereit
et al., 2022a).
The representation of wind farms in NWP and climate models has evolved significantly over the past two decades, mov-
ing from surface-based approximations to more physically realistic parameterizations. In the early 2000s, wind farms were
incorporated into NWP and climate models through enhanced surface roughness ($z_0$) or drag (Ivanova and Nadyozhina, 2000;
Malyshev et al., 2003; Keith et al., 2004; Wang and Prinn, 2010). However, this approach led to exaggerated surface fluxes, tur-
bulence kinetic energy (TKE), and wind speed deficits (Fitch et al., 2013b). A key conceptual shift came with Baidya Roy et al.
(2004), who proposed that wind farms act as an elevated momentum sink and TKE source within the rotor layer rather than
at the surface. The elevated momentum sink and TKE source framework remains the conceptual foundation for most WFPs
in use today, and for several subsequent advances (Fitch et al., 2012; Adams and Keith, 2013; Boettcher et al., 2015; Abkar
and Porté-Agel, 2015; Volker et al., 2015; Pan and Archer, 2018; Redfern et al., 2019; Ma et al., 2022). For a comprehensive
overview of WFP developments, see the review by Fischereit et al. (2022a).
One of the key advances in WFPs was the modification of the TKE source term, initially treated as a constant (Baidya
Roy et al., 2004). Blahak et al. (2010) proposed linking the added TKE to the energy extracted by the turbines via the power
coefficient ($C_P$), and Fitch et al. (2012) introduced the formulation $C_{\mathrm{TKE}} = C_T - C_P$, where $C_T$ is the thrust coefficient.
Later, Archer et al. (2020) suggested scaling $C_{\mathrm{TKE}}$ with a TKE factor ($\alpha$) of 0.25 to avoid exaggerated TKE values. Even
though some studies find better performance using $\alpha = 1.00$ instead (Larsén and Fischereit, 2021), the impact of changing $\alpha$
varies spatially (Rosencrans et al., 2024; Ali et al., 2023; García-Santiago et al., 2024). Most recently, large-eddy simulation
(LES)-based formulations have been proposed to further improve the TKE source term (Khanjari et al., 2025).
Regarding the WFPs and the representation of atmosphere–wind farm interactions, it is useful to distinguish between grid-
unresolved and grid-resolved processes. Unresolved processes include the momentum sink term, intra-grid-cell interactions
between turbines (Abkar and Porté-Agel, 2015; Pan and Archer, 2018; Ma et al., 2022), local wake expansion (Volker et al.,
2015), and wake recovery occurring within the same grid cell. The momentum sink term creates a wind speed deficit (wake) as
a grid-unresolved process. However, momentum recovery into the generated wake is a spatial process that occurs over several
grid cells and depends on spatial gradients of wind speed, direction and TKE. Wake recovery is therefore a grid-resolved
process. This grid-resolved wake recovery depends on multiple factors. Several studies have shown sensitivity to horizontal
grid resolution (Mangara et al., 2019; Siedersleben et al., 2020; Tomaszewski and Lundquist, 2020; Peña et al., 2022; Sanchez
Gomez et al., 2024) and vertical resolution (Vanderwende et al., 2016; Mangara et al., 2019; Tomaszewski and Lundquist,
2020; Siedersleben et al., 2020). The planetary boundary layer (PBL) scheme is also influential (Rybchuk et al., 2022; Agarwal





et al., 2025), as is the tuning of the TKE addition factor $\alpha$ (Archer et al., 2020; Siedersleben et al., 2020; Tomaszewski and
Lundquist, 2020; Larsén and Fischereit, 2021; Sanchez Gomez et al., 2024). In addition, atmospheric stability plays a key role in
modulating wake recovery and has been highlighted in several recent works (Vanderwende et al., 2016; Lundquist et al., 2019;
Rosencrans et al., 2024; García-Santiago et al., 2024; Quint et al., 2025). Performance also varies between different WFPs.
For instance, the Explicit Wake Parameterization (EWP) generally produces shorter wakes than the Fitch scheme (Shepherd
et al., 2020; Pryor et al., 2020; Larsén and Fischereit, 2021; Fischereit et al., 2022b; García-Santiago et al., 2024; Pryor and
Barthelmie, 2024).
The underestimation of wake recovery in NWP-WFP simulations is related to grid-resolved processes. A conceptual descrip-
tion of the generation and recovery of the wake in NWP-WFP, in comparison with LES, is provided in Fig. 1. Two downstream
cells are used because the added TKE in the LES usually reaches its maximum in the first downstream cell (Fig. 1e), whereas
the added TKE maximum occurs at the turbine grid cell for the NWP-WFP (Fig. 1d). Thus, wake recovery and TKE decay
are assessed here as streamwise changes between two consecutive downstream cells (Figs. 1b and c; e and f). The NWP-WFP
$\Delta WS$ barely changes between panels (Figs. 1b and c), whereas the LES displays recovery. On the other hand, the $\Delta TKE$ de-
cays too fast for the NWP-WFP compared with the LES between panels (Figs. 1e and f). Because the slow wake recovery and
the fast TKE decay in the NWP-WFP occur over downstream cells without turbines, they are not caused by the WFP. Multiple
NWP-WFP studies report good agreement of the wind speed deficit with LES within turbine or farm grid cells (Abkar and
Porté-Agel, 2015; Vanderwende et al., 2016; Peña et al., 2022; García-Santiago et al., 2024), suggesting that the momentum
sink term (i.e., the grid-unresolved component) creates a wind speed deficit consistent with the LES (Fig. 1a). However, after
the generation of the wind speed deficit, the downstream wind speed profiles often remain unchanged in the NWP compared to
the LES (Figs. 1b and c). This discrepancy in wake recovery between NWP-WFP and LES persists across neutral, convective,
and stable atmospheric stability regimes (García-Santiago et al., 2024). Ultimately, this slow recovery in NWP-WFP simu-
lations likely contributes to the overestimation of power losses often reported when using WFPs (Lee and Lundquist, 2017;
Montavon et al., 2024). Thus, the evidence of a well-functioning momentum sink term combined with the insufficient wake
recovery downstream of turbine grid cells suggests the problem lies in the grid-resolved wake recovery process.
Another issue, less frequently discussed, is the rapid decay of farm-induced TKE in NWP-WFP simulations. While less
obvious, this problem is evident when closely examining the spatial evolution of TKE profiles downstream of turbine grid
cells. In NWP-WFP simulations, TKE decays much more rapidly than in LES, as conceptually illustrated in Figs. 1e and f,
and supported by evidence from the literature (Figs. 5a–d in Vanderwende et al. (2016); Figs. 4 and 9 in García-Santiago et al.
(2024); Figs. 11-13 in Peña et al. (2022)). As more studies focus on wake recovery in large farms, this issue is becoming more
visible. For example, Rosencrans et al. (2024) found that the amount of TKE added by the farm influences near-farm wake
deficits but not the overall wake length. Similarly, Sanchez Gomez et al. (2024) noted that wake losses at an offshore wind farm
located approximately 15 km downstream of another farm were largely unaffected by the TKE factor. However, omitting the
TKE factor ($\alpha = 0$) resulted in the largest biases when compared with SCADA data. Furthermore, aircraft observations have
shown that TKE is overestimated in the first half of the farm by NWP-WFP, while in reality, TKE peaks further downstream
(Siedersleben et al., 2020).



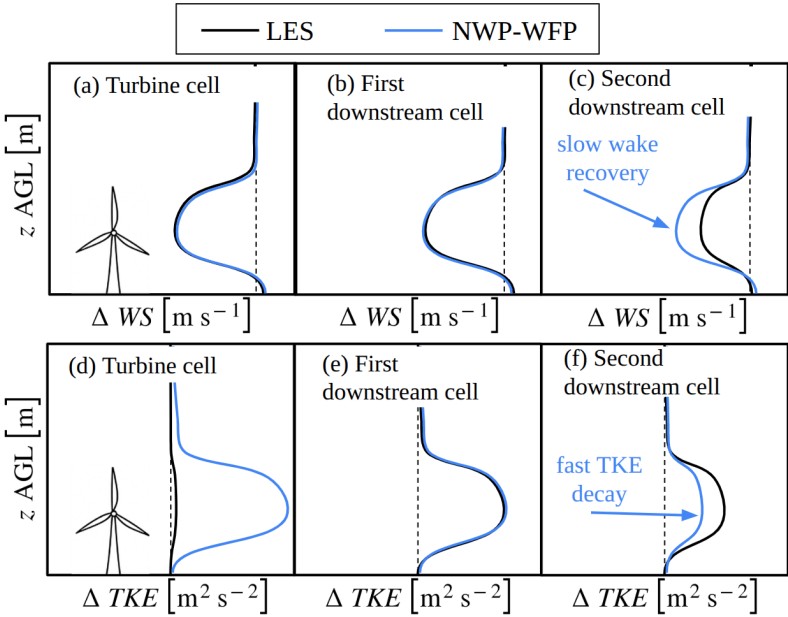

**Figure 1.** Conceptual description of wind turbine wake generation and recovery in a NWP-WFP simulation provided as a summary of the literature (Abkar and Porté-Agel, 2015; Vanderwende et al., 2016; Peña et al., 2022; García-Santiago et al., 2024). Time-averaged vertical profiles of wind speed deficit ($\Delta WS$, a–c) and turbine-added TKE ($\Delta TKE$, d–f) within the turbine grid cell (a, d), for the first (b, e) and second (c, f) downstream cells in the NWP without turbines. The wind speed deficit is defined as the difference between simulations with and without turbines ($\Delta WS = WS_T - WS_{NT}$), respectively, yielding negative values in most of the wake. The turbine-added TKE ($\Delta TKE = TKE_T - TKE_{NT}$) is similarly defined but typically yields positive values.

These two interrelated issues—the underestimated wake recovery and the fast decay of the farm-added TKE—need a closer
examination. While introduced here as two separate problems, turbulent mixing is a key driver for wake recovery (Vermeer
et al., 2003; Abkar and Porté-Agel, 2015; van der Laan et al., 2023). Unrealistic decay of the TKE can slow down wake
recovery in NWP-WFP simulations. Therefore, these two issues raise a few important questions. For instance, why does the
farm-added TKE decay more quickly in NWP-WFP simulations than in the LES? How much does this rapid decay contribute
to the underestimated wake recovery? What other physical or numerical factors are involved? To address these questions,
we evaluate the representation of wind farm wake recovery in the NWP-WFP approach. Specifically, we aim to quantify the
magnitude and spatial variability of the recovery process and identify the mechanisms behind the underestimation. This effort
requires a large-domain LES to evaluate wake recovery within, immediately downstream and in the far wake of the wind farm.
To our knowledge, there are no NWP-WFP vs. LES comparisons over downstream distances of ∼40 km.
The remainder of this article is structured as follows. Section 2 describes the numerical setups used in this study, including
the WRF simulations with the Fitch et al. (2012) wind farm parameterization (NWP-WFP) and the reference LES with actuator
disks (Kasper et al., 2024). The idealized simulations focus on a 600 MW offshore wind farm with an aligned layout, subjected



to westerly flow under near-neutral atmospheric conditions. In Section 3, we compare the undisturbed inflow conditions across
models to ensure consistency. We then assess wind farm performance and analyze wake recovery in the streamwise direction.
Section 4 discusses in depth the two main limitations of wake recovery in NWP-WFP: the fast decay of farm-added TKE and
the under-resolved gradients in the wind farm wake. Finally, Section 5 summarizes the key findings, with particular emphasis
on how these limitations impact the simulation of wind farm cluster wakes with NWP-WFP models.

## 2    Methods

To benchmark the NWP simulations with the WFP, we compare the results against LES by Kasper et al. (2024), who studied
atmosphere–wind farm interactions under idealized conditions. Specifically, we consider the Barotropic (BT) case of their
study, which here serves as the reference case that the mesoscale NWP simulations are designed to replicate. The simulation
setup for this LES is briefly covered in Section 2.1, while a thorough discussion on the numerical framework can be found in
Kasper et al. (2024). The subsequent Section 2.2 details the NWP-WFP framework, and the adaptations required to represent
the LES conditions within a mesoscale model.

### 2.1    Setup for the LES

The reference simulation uses a modified version of the LES code developed by Albertson and Parlange (1999), later validated
in Gadde et al. (2021). The model solves the incompressible filtered mass conservation, Navier–Stokes, and potential temper-
ature transport equations. The subgrid-scale stresses and heat fluxes are modeled using the anisotropic minimum dissipation
(AMD) scheme (Rozema et al., 2015; Abkar et al., 2016a), suitable for stratified boundary layers and wind farm wakes.
In the horizontal directions, the code employs pseudo-spectral differentiation and periodic boundary conditions, while
second-order finite differencing is used in the vertical direction. A free-slip boundary condition is imposed at the top, with
a Rayleigh damping layer to minimize gravity wave reflections. At the surface, a zero heat flux condition enforces neutral
stratification, and shear stresses are parameterized using Monin–Obukhov similarity theory. Time integration uses a third-
order Adams–Bashforth scheme. The simulation domain spans $102.4 \times 10.24 \times 10$ km$^3$ in the streamwise, spanwise, and
vertical directions, respectively. It is discretized using a rectilinear grid that consists of $2048 \times 512 \times 384$ points, with
horizontal resolutions of $\Delta x = 50$ m and $\Delta y = 20$ m. In the vertical, the resolution is uniform with $\Delta z = 10$ m up to
$z_u = 1.5$ km above ground level (AGL), and stretched above using a hyperbolic tangent profile up to a maximum spacing
of 62 m.
The simulated atmosphere represents an idealized offshore environment based on North Sea conditions. The geostrophic
wind vector is prescribed with a magnitude $|\mathbf{U}_g| \approx 10$ m s$^{-1}$, and the Coriolis frequency equals $f_c = 1.159 \times 10^{-4}$ s$^{-1}$.
The surface roughness is set to $z_0 = 0.002$ m, typical for open-sea conditions. The background stratification is neutral up
to 1 km AGL, with a potential temperature of $\theta = 286$ K. The boundary layer is capped by a 3 K inversion over 200 m,
followed by a free-atmosphere lapse rate of 5 K km$^{-1}$.



Wind turbines are represented using the actuator disk model (Jimenez et al., 2008; Calaf et al., 2010). The simulated wind
farm consists of ten rows and six columns of turbines in an aligned configuration, spaced by $s_x = 7D$ and $s_y = 5D$ in stream-
wise and spanwise direction, respectively. The turbine diameter equals $D = 178$ m and the hub-height $z_h = 119$ m AGL, corre-
sponding to the DTU 10-MW reference turbine (Bak et al., 2013). A uniform thrust coefficient $C_T = 0.75$ is used, and turbines
yaw to face the local incoming wind.
The LES is run for a total of 11 hours and comprises two stages. First, a 7-hour spin-up simulation is run on a coarser grid with
a resolution of $2\Delta_x \times 2\Delta_y \times \Delta_z$. Second, once the boundary layer reaches quasi-equilibrium (the mean wind and turbulence
statistics display small variations over time), it is interpolated to the full resolution and the LES proceeds as a precursor-
successor simulation (Stevens et al., 2014). The precursor provides realistic turbulent inflow to the successor domain containing
the wind farm, preventing contamination of the inflow by remnants of the wakes via the periodic boundary conditions. Statistics
are then collected over the final 3 hours of the simulation.

## 148    2.2    Setup for the NWP-WFP simulations

The NWP simulations use the Advanced-Research WRF model version 4.4 (Skamarock et al., 2019), which solves the com-
pressible Euler equations in three spatial dimensions and time. The solver uses a time-split integration scheme. The low-
frequency modes are integrated with a third-order Runge–Kutta scheme, whereas higher-frequency acoustic modes are inte-
grated over smaller time steps. Advective terms are discretized using a fifth-order scheme in the horizontal and a third-order
scheme in the vertical directions. The model applies Arakawa C-grid staggering in the horizontal direction, with a hydrostatic
pressure-based coordinate in the vertical direction.
Idealized simulations are employed, omitting cloud microphysics, radiation, moisture, and surface heterogeneity, which are
standard simplifications in studies targeting canonical boundary layers over flat terrain (Fitch et al., 2012; Vanderwende et al.,
2016; Volker et al., 2015; Peña et al., 2022; García-Santiago et al., 2024). In the multiscale framework of the WRF model,
turbulence is entirely parameterized by the PBL scheme for mesoscale grid spacings ($\Delta X \sim 1$ km).
A two-domain nesting configuration is adopted (Fig. 2), with one-way coupling from a parent domain to an inner nest.
The outer domain generates nearly steady boundary-layer inflow characteristics, which are passed to the nested domain. Both
domains use a constant horizontal resolution ($\Delta X = \Delta Y$), hereafter referred to simply as $\Delta X$, as specified in Table 1, ranging
from 346 m to 1246 m for the sets of simulations considered here. Simulations with the finest horizontal grid resolutions
($\Delta X = 2D$, or 346 m, DX2D and DX2DTKE100) use a coarser resolution ($\Delta X = 5D$, 890 m) for the outer domain, with a
parent grid aspect ratio of 3. Vertically, a fine 10-m resolution is used below 400 m AGL for all domains, resolving the turbine
rotor layer with multiple grid points. The computational domain is larger than in the LES to further minimize the influence of
lateral boundaries, as the added computational cost is relatively small for the NWP-WFP simulations. For most cases (except
DX2D and DX2DTKE100) the innermost domain spans approximately 188 km streamwise and 63 km spanwise. In the finest-
resolution cases (DX2D and DX2DTKE100), it spans about 100 km and 40 km in the streamwise and spanwise directions,
respectively. All domains extend 5 km vertically.





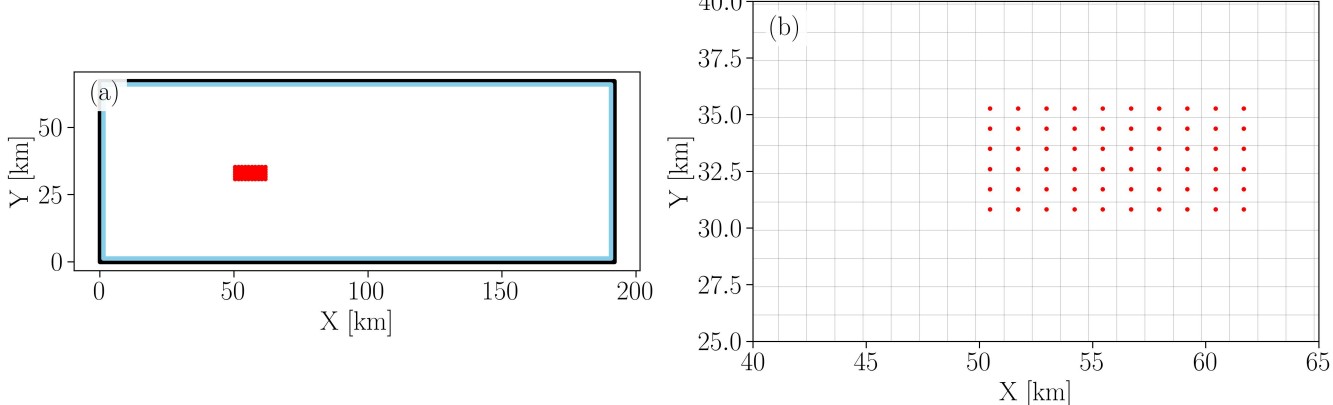

**Figure 2.** Two computational domains are used in the WRF-WFP simulations, illustrated here for the baseline (BASE) case (a). The outer parent domain (D1, black rectangle), which uses periodic lateral boundary conditions and does not include a wind farm, provides inflow to the inner nested domain (D2, blue rectangle) via one-way nesting. The inner domain includes the wind farm, with turbines represented as red dots. In the BASE setup shown in panel (b), each grid cell contains one turbine, except for the cells between $Y =$ 32–34 km, which contain two.

**Table 1.** Summary of the NWP-WFP simulations, their subsets and key parameters: horizontal grid resolution ($\Delta X$), TKE addition coefficient ($\alpha$) and turbine thrust coefficient ($C_T$).

| Case | Subset | $\Delta X$ [m] | $\alpha$ | $C_T$ |
|---|---|---|---|---|
| BASE | Baseline | 1246 ($7D$) | 0.25 | 0.75 |
| TKE000 | TKE | 1246 ($7D$) | 0.00 | 0.75 |
| TKE050 | TKE | 1246 ($7D$) | 0.50 | 0.75 |
| TKE075 | TKE | 1246 ($7D$) | 0.75 | 0.75 |
| TKE100 | TKE | 1246 ($7D$) | 1.00 | 0.75 |
| DX5D | Grid | 890 ($5D$) | 0.25 | 0.75 |
| DX2D | Grid | 346 ($2D$) | 0.25 | 0.75 |
| DX2DTKE100 | Grid | 346 ($2D$) | 1.00 | 0.75 |
| CT050 | Thrust | 1246 ($7D$) | 0.25 | 0.50 |
| CT081 | Thrust | 1246 ($7D$) | 0.25 | 0.81 |
| CTTC | Thrust | 1246 ($7D$) | 0.25 | Thrust Curve (TC) |

The wind farm is represented using the Fitch WFP (Fitch et al., 2012), with TKE advection enabled (Archer et al., 2020) and
axial induction modified following Vollmer et al. (2024). The wind farm consists of 60 DTU 10-MW turbines (Bak et al., 2013),
configured identically to Kasper et al. (2024). Each turbine has a rotor diameter $D = 178$ m and a hub height $z_h = 119$ m AGL.
A constant thrust coefficient of $C_T = 0.75$ is used in all cases, except for those included in the $C_T$ sensitivity study. Details
on the turbine model and the sensitivity analysis are provided in Appendix A. The wind farm occupies a region starting
around $X = 50$ km and $Y = 30$ km from the western and southern boundaries. The wind farm spans 11 km streamwise and

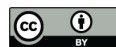



4.4 km spanwise, with turbine spacing of $7D$ (1246 m) in the streamwise and $5D$ (890 m) in the spanwise directions. Uniform
turbine spacing in the coarse NWP-WFP grid is only achieved when grid spacing matches the physical turbine spacing. For
instance, the BASE case has a uniform horizontal spacing which ensures a uniform representation of turbine spacing in the
streamwise direction ($\Delta X = S_x = 7D$), but not in the spanwise direction where turbines are more closely spaced ($\Delta X = 7D$
but $S_y = 5D$).
Surface boundary conditions assume flat, homogeneous terrain with roughness length $z_0 = 0.002$ m and zero surface heat
flux (neutral stability). The surface-layer scheme applies Monin-Okukhov Similarity Theory (MOST)-based drag using near-
surface wind speeds. The Coriolis parameter corresponds to a latitude of 52.65°. The upper boundary enforces zero vertical
velocity and free-slip horizontal flow. A Rayleigh damping layer (coefficient $0.2$ s$^{-1}$) occupies the top 2 km of the domain to
absorb gravity waves.
Initial conditions prescribe a uniform, westerly flow from 282° at 10.93 m s$^{-1}$. The initial potential temperature is uniform
at 286 K below 1000 m AGL, then increases with lapse rates of 15 K km$^{-1}$ (1000–1200 m AGL) and 5 K km$^{-1}$ (above 1200 m
to the 5-km AGL domain top) to match the LES. Simulations run for 27 hours, with the final hour used for analysis after a
26-hour spin-up. The initial forcing is fine-tuned to ensure post-spin-up conditions match target values, a standard approach
in idealized WRF setups (Mirocha et al., 2018; Peña et al., 2021; Hsieh et al., 2025). Inertial oscillations driven by Coriolis
effects also modulate boundary-layer winds during spin-up. The chosen spin-up time ensures minimal residual oscillations in
the analysis period.
Three simulation sets and a baseline case are run (Table 1). The baseline (BASE) case uses a horizontal resolution equal to the
turbine streamwise spacing ($7D$) to avoid subgrid wake effects from multiple turbines in one cell. It applies a thrust coefficient
of $C_T = 0.75$ (such as in (Kasper et al., 2024)) and a TKE addition coefficient of $\alpha = 0.25$ (Archer et al., 2020). The TKE
subset includes $\alpha = 0.00, 0.50, 0.75$ and $1.00$. The grid subset varies $\Delta X$ between 346 m ($2D$) and 890 m ($5D$). The thrust
subset applies constant $C_T$ values of 0.50 and 0.81, and also includes a wind-speed-dependent thrust case (see Appendix A).
This subset serves to assess the sensitivity of the wake generation and recovery relative to the choice of $C_T = 0.75$ in the LES.
The high-resolution cases DX2D and DX2DTKE100 are used to assess the impact of sharper wake gradients on power and
recovery, acknowledging limitations of the application of traditional PBL schemes at sub-kilometer resolutions due to the *terra*
*incognita* (Wyngaard, 2004; Rai et al., 2019; Haupt et al., 2019, 2023).

## 3   Results

### 3.1   Undisturbed inflow profiles

Before delving into modeling differences and similarities between NWP-WFP and LES in terms of power production and
wake recovery, it is necessary to evaluate inflow variability to ensure inflow profiles are sufficiently similar so that downstream
differences are not attributed to inflow variability (Peña et al., 2022; Hsieh et al., 2025). There is excellent agreement between
NWP-WFP and LES inflow profiles for all variables (Fig. 3). The differences in hub-height ($WS_{119}$) and rotor-averaged ($WS_r$)
wind speed compared to the LES are approximately 0.03 and $-0.01$ m s$^{-1}$, respectively, as shown in Table 2.






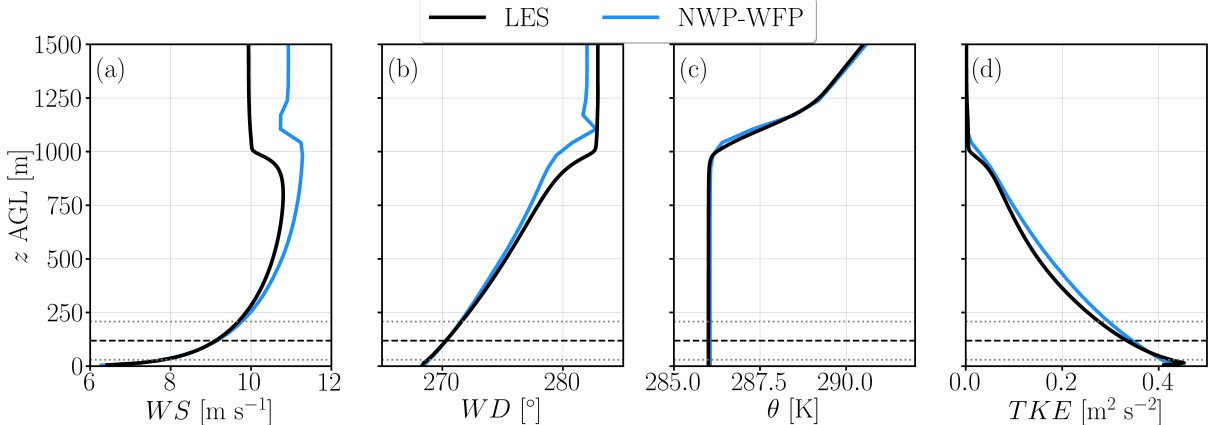

**Figure 3.** Time- and horizontally-averaged profiles of wind speed (a), wind direction (b), potential temperature (c), and TKE (d) during the analysis window for LES (black line) and NWP-WFP (blue line) in the simulation without turbines. The horizontal dashed and dotted gray lines denote rotor hub height and bottom/top tips, respectively.

Not only do the hub-height and rotor-averaged values match well, but so do the wind shear ($\alpha_r$) and veer ($\beta_r$) across the
rotor (Figs. 3a, b). This is important because under near-neutral conditions, turbulence production is governed primarily by
mechanical shear, both in wind speed and direction (Stull, 1988). The fact that similar wind speed and direction profiles result
in similar TKE profiles (Fig. 3d) reinforces the physical consistency between the models. The NWP-WFP wind speed profile
exhibits slightly greater shear between 300 and 800 m AGL, which leads to marginally higher TKE values in that layer. Surface
boundary conditions are also consistent, as indicated by the good agreement in friction velocity ($u_\star$, Table 2).

**Table 2.** Time-averaged inflow properties from the precursor simulations and their differences. Subscripts 119 and $r$ denote hub-height (in m AGL) and rotor-averaged values, respectively. For wind shear ($\Delta WS_r$) and veer ($\Delta WD_r$), values represent the difference between the top and bottom rotor tips.

| Source | $WS_{119}$ | $WD_{119}$ | $TKE_{119}$ | $WS_r$ | $WD_r$ | $TKE_r$ | $u_\star$ | $\Delta WS_r$ | $\Delta WD_r$ |
|---|---|---|---|---|---|---|---|---|---|
| | [m s$^{-1}$] | [°] | [m$^2$ s$^{-2}$] | [m s$^{-1}$] | [°] | [m$^2$ s$^{-2}$] | [m s$^{-1}$] | [m s$^{-1}$] | [°] |
| NWP-WFP | 9.15 | 270.3 | 0.35 | 9.02 | 270.3 | 0.35 | 0.32 | 1.98 | 2.58 |
| LES | 9.12 | 270.3 | 0.33 | 9.03 | 270.3 | 0.34 | 0.33 | 1.87 | 2.81 |
| Difference | 0.03 | 0.0 | 0.01 | -0.01 | 0.0 | 0.01 | -0.01 | 0.12 | -0.23 |

## 215   3.2   Row-averaged streamwise power patterns

This section evaluates streamwise changes in row-averaged power (i.e., averaged across turbines within the same row) for the
TKE (Fig. 4a) and grid resolution (Fig. 4b) simulation subsets. These results reflect both the magnitude of wake effects on
downstream turbines and the ability of the waked flow to recover momentum within the wind farm.

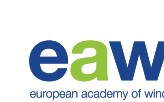
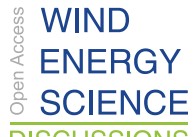

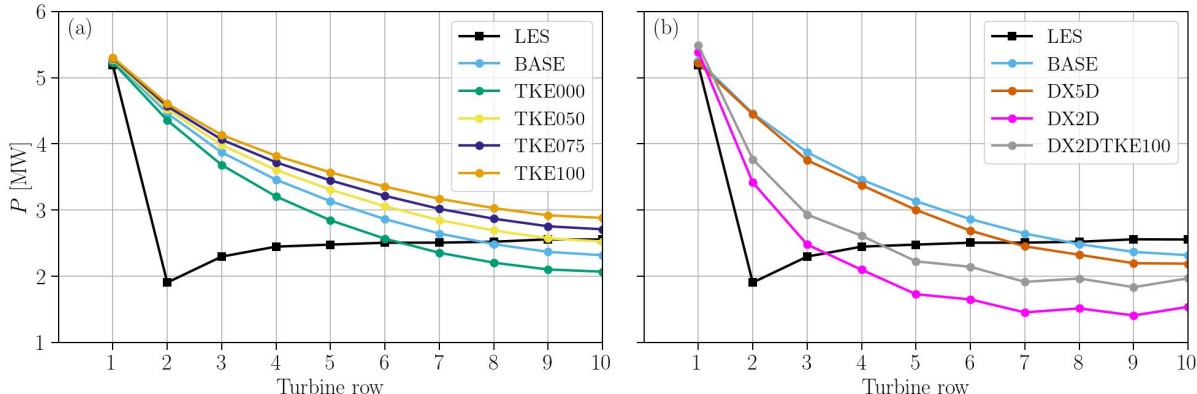

**Figure 4.** Row-averaged power in the NWP-WFP set of simulations with different wind farm added TKE (a) and grid resolutions (b) compared with the LES.

Compared to the LES, there is excellent agreement in the first-row averaged power output ($\sim$5 MW; Fig. 4a), which stems from the matched inflow profiles (Fig. 3) and the implementation of the WFP correction (Vollmer et al., 2024). This agreement indicates that turbine power is consistently modeled across both simulation frameworks, enabling a meaningful evaluation of model-specific biases. Additionally, increasing the farm-added TKE enhances row-averaged power output by promoting more efficient wake recovery via momentum transport into the farm.

In the LES, the power drops sharply in the second row (to $\sim$2 MW) and is followed by a gradual power recovery downstream (Figs. 4a,b), as is often observed in LES studies of dense offshore wind farms with an aligned layout (Allaerts and Meyers, 2017; Stevens and Meneveau, 2017; Stieren and Stevens, 2022; Stipa et al., 2024). For staggered layouts, the power decrease is smaller because the effective streamwise distance between turbine rows doubles compared to the aligned layout (Stieren and Stevens, 2022). None of the NWP-WFP simulations capture the sharp drop in power, instead displaying a more gradual pattern of power decrease. Only the DX2D and DX2DTKE100 cases (Fig. 4b) show some agreement with the LES owing to their finer spatial resolution ($\Delta X = 2D$). Coarser-resolution cases, regardless of the added TKE (Figs. 4a), tend to overestimate power in the upstream half of the farm. This power overestimation results from weaker wake losses, as the coarse NWP-WFP grid ($\Delta X = 1$ km) dilutes wake structures spatially. This limitation has been previously documented (Archer et al., 2020; Sanchez Gomez et al., 2024).

At finer resolution (cases DX2D and DX2DTKE100), wakes become sharper and induce more realistic power deficits in turbine rows 2–4 (Fig. 4b). The absence of power recovery beyond the second row in the NWP-WFP simulations, compared to the LES (Figs. 4a,b), likely stems from insufficient wake recovery, an aspect discussed further in Section 3.4. Consequently, power performance patterns produced by NWP-WFP simulations may differ significantly from those in the LES. This discrepancy should be considered when employing NWP-WFP for intra-farm power performance assessments.





## 3.3 Wind farm wake flow field and spatial average

In this section, we compare the representations of hub-height wind speed and TKE by the LES and NWP-WFP simulations. The goal is to quantify the differences in wind farm wake structure and recovery between the LES, spatially averaged to the grid of the BASE case (coarsened LES), and selected NWP-WFP cases (Fig. 5). The coarsened LES result is obtained by spatially averaging the full-resolution LES data located within individual grid cells of the BASE case, enabling a direct comparison between the two (Figs. 5k,l), as in previous studies (Vanderwende et al., 2016; Peña et al., 2022; García-Santiago et al., 2024). The origin of the coordinate system is shifted to the southwest corner of the wind farm. Throughout the text, we divide the wind farm wake into three distinct regions: the intra-farm wake ($0 \text{ km} < X < 12 \text{ km}$), the near-farm wake ($12 \text{ km} < X < 15 \text{ km}$), and the far wake of the farm ($X > 15 \text{ km}$). The wind farm exit separates the intra-farm and near-farm wake regions. The thick black rectangle shows the wind farm perimeter as defined by the WFP. In case DX2D (Fig. 5i), the finer resolution results in a smaller represented perimeter compared to the BASE case.

The full-resolution LES (Figs. 5a,b) exhibits sharp streaks of alternating low and high wind speed and TKE due to the aligned turbine layout (Stieren and Stevens, 2022; Stipa et al., 2024). These streaks extend approximately 3–5 km downstream before merging into a single and broader structure (Fig. 5a). In contrast, the coarsened LES aggregates and smooths out these microscale variabilities, producing more homogeneous flow and turbulence fields within the farm. In the far wake region ($X > 15 \text{ km}$), the LES and coarsened LES fields become similar, as turbulent mixing has reduced the spatial gradients in wind speed and TKE.

A persistent feature in the coarsened LES, BASE and TKE100 (Figs. 5c,e,g) cases is a narrow band of reduced wind speed around $Y = 2.5 \text{ km}$. This more intense wake results from the $Y$-direction turbine spacing ($s_y = 5D$) being finer than the grid spacing ($\Delta X = 7D$), such that two turbine columns fall within a single grid cell. This aggregation produces a locally stronger momentum sink and TKE source (Fitch et al., 2012).

Although the total momentum sink in DX2D is comparable to BASE and TKE100, the local wind speed deficits are larger due to the finer resolution, which produces more concentrated wakes (Figs. 5i). This localization of the wind speed deficits explains the improved agreement with the LES regarding the second-row power losses for cases DX2D and DX2DTKE100 (Fig. 4b).

The wind speed ($\Delta WS_{119}$) and TKE ($\Delta TKE_{119}$) differences, calculated as the BASE case minus the coarsened LES (Figs. 5k,l), are small upstream and beside the wind farm. Faint blue bands of negative $\Delta WS_{119}$ appear outside the spanwise averaging region, caused by stronger acceleration around the wind farm in the LES. Within the farm, $\Delta WS_{119}$ transitions from slightly positive in the first few rows, to near-zero at row 4, and then to negative further downstream. The strongest negative wind speed differences occur in the near-farm wake ($12 \text{ km} < X < 15 \text{ km}$), followed by a modest recovery, but remain between $-0.5$ and $-1 \text{ m s}^{-1}$ in the far wake of the farm ($X > 20 \text{ km}$). For $\Delta TKE_{119}$, the BASE case overestimates TKE compared to the LES in the first turbine row and underestimates it in rows 3 and further downstream. In the near wake ($12 \text{ km} < X < 15 \text{ km}$), the TKE remains underestimated but recovers further downstream ($X > 20 \text{ km}$) where the agreement with the LES is excellent.



**Figure 5.** Time-averaged wind speed ($WS_{119}$) and TKE ($TKE_{119}$) at hub height for the LES (a, b), coarsened LES (c, d), NWP-WFP cases BASE (e, f), TKE100 (g, h), and DX2D (i, j). The last row shows the difference between the BASE case and the coarsened LES (k, l). The black rectangles indicate the wind farm perimeter as represented in the NWP-WFP simulation. Horizontal dashed lines mark the spanwise extent of the region used for averaging.





## 3.4 Intra-farm, near-farm and far wake recovery

Here, we assess how the streamwise evolution of the wind farm wake is affected by changing the TKE coefficient (Fig. 6) and the grid resolution (Fig. 7). The time-averaged wind speed and TKE are evaluated at hub height and averaged in the spanwise direction within the bounds ($Y = -0.923$ and $5.307$ km) of the horizontal dashed lines represented in Fig. 5, the lateral extent of the wind farm in the reference BASE case. Furthermore, to mitigate averaging errors from the coarse mesoscale resolution near the spanwise boundaries, all results are first interpolated to a higher-resolution grid ($\Delta X = 50$ m).

**Figure 6.** Streamwise variation of spanwise averages over the wind farm area of hub-height wind speed (a), wind direction (b), TKE (c), and streamwise gradient of wind speed (d) for the LES at full resolution, coarsened LES, and NWP-WFP cases with different $\alpha$. The colored background areas indicate the streamwise extent of the intra-farm wake (gray), near-farm wake (green), and far wake of the farm (blue).



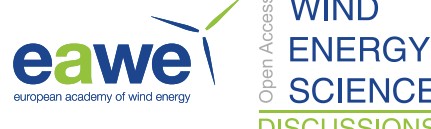

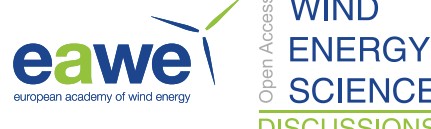

**Figure 7.** Streamwise variation of spanwise averages over the wind farm area of hub-height wind speed (a), wind direction (b), TKE (c), and streamwise gradient of wind speed (d) for the LES at full resolution, coarsened LES, and NWP-WFP cases with different grid resolution. The colored background areas indicate the streamwise extent of the intra-farm wake (gray), near-farm wake (green), and far wake of the farm (blue).

The NWP-WFP simulations slightly underestimate the wind speed deficit in the first three turbine rows of the wind farm (leading to overestimated power production, Fig. 4) but overestimate the deficit further downstream (Figs. 6a and 7a). Increasing the added TKE has little impact on the first few turbine rows, because momentum entrainment builds up progressively row by row, with its cumulative effects most evident at the wind farm exit. Notably, the cases with $\alpha$ values of 75% and 100% show clear improvement at the wind farm exit and in the near-wake region ($X < 15$ km). Differences in wind speed between the NWP-WFP cases diminish in the far wake (Figs. 6a and 7a). Overall, regardless of the TKE coefficient, all the mesoscale simulations display a consistent negative wind speed bias ranging from approximately $-0.7$ m s$^{-1}$ (TKE000) to $-1.3$ m s$^{-1}$





(TKE100) in the near-farm wake at $X = 15$ km. The bias decreases to approximately $-0.6$ m s$^{-1}$ in the far wake ($X = 50$ km)
for all cases (Fig. 6a) compared to the LES.
The grid resolution exerts a stronger impact on improving the bias. The bias improves considerably for the two finest resolu-
tion cases DX2D and DX2DTKE100 (horizontal resolution of $2D$), with wind speed deficits of $-0.3$ m s$^{-1}$ (DX2DTKE100)
and $-0.5$ m s$^{-1}$ (DX2D) in the near-farm wake at $X = 15$ km (Fig. 7a). In the far wake at $X = 50$ km, the bias of approx-
imately $-0.4$ m s$^{-1}$ for the DX2D and DX2DTKE100 remains relatively similar to the values in the near-farm wake (from
$-0.3$ to $-0.5$ m s$^{-1}$). However, the BASE case ($7D$ or 1246 m resolution) and the DX5D case ($5D$ or 890 m resolution) yield
very similar results, suggesting that the modest increase in mesoscale grid resolution does not noticeably affect the wake.
The streamwise gradient of wind speed ($\Delta WS_{119}/\Delta X$) quantifies wake recovery (Fig. 6d). Among the cases varying the
TKE coefficient, TKE100 most closely approximates the LES within the farm area. In the far wake ($X > 20$km), streamwise
derivatives across all cases converge and agree well with the LES, despite the bias in absolute wind speed. However, the
greatest divergence occurs in the near-farm wake ($12 < X < 15$ km), where the NWP-WFP simulations all fail to capture a
peak in recovery rate (Fig. 6d). Thus, that wake recovery in the LES is faster immediately after the wind farm exit, which is also
noticeable in the shape of the streamwise wind speed curvature (Fig. 6a). Even case TKE100, which most closely approximates
the LES wind speed in the near-farm wake, underestimates the wake recovery rate downstream. This discrepancy suggests that
a key mechanism for wake recovery is not adequately captured in the NWP-WFP simulations in the near-farm wake.
Simulations with the finest grid resolution (DX2D and DX2DTKE100) improve the representation of wind speed in the
intra-farm and near-farm wake regions (Figs. 7a,d). This improvement is first attributed to the more realistic predictions of
turbine power in these cases (Fig. 4b), which imply a more accurate extraction of momentum by the wind farm. As a result,
the generated wake exhibits a wind speed deficit that more closely matches the LES near the wind farm exit. Secondly, the
near-farm wake recovers more quickly than in the coarser-resolution cases, as evident in both the wind speed (Fig. 7a) and the
streamwise gradients in wind speed (Fig. 7d), which now reproduce the peak displayed by the LES. Finally, adding more TKE
($\alpha = 100\%$) in the high-resolution simulation further improves wake generation and recovery, yielding the best agreement with
the LES.
In the LES, TKE builds up gradually, row by row, whereas in the NWP-WFP cases with a TKE source ($\alpha > 0$), the TKE
peaks near the first few turbine rows and then decreases monotonically downstream (Fig. 6c). The TKE remains nearly constant
within the farm in case BASE, and is small throughout in case TKE000, which lacks an explicit TKE source. Interestingly,
cases with $\alpha$ of 25% and 50% better match the TKE levels near the first turbine rows, consistent with results from Archer
et al. (2020), while those with higher $\alpha$ (75%, 100%) better match the TKE in the latter half of the farm, in line with other
findings (Larsén and Fischereit, 2021; Ali et al., 2023). This variability in the literature seems to stem in part from where the
comparison is made: within, near or further downstream of the wind farm, as shown here.
A key insight is that turbine-added TKE cannot accumulate properly in the NWP-WFP simulations, even if its magnitude
matches or exceeds that of the LES in the first few rows of turbines. The root problem lies not only in the amount of turbine-
added TKE, but in its rapid decay. A large TKE dissipation rate in the NWP simulations appears to inhibit the streamwise
accumulation of turbine-added TKE that is evident in the LES. Supporting this interpretation, the much faster decay of turbine-





added TKE in the NWP-WFP simulations in the near-farm wake (Fig. 6c), compared to the LES, provides a clear signal of
what initially appears to be exaggerated TKE dissipation. However, in Section 4.1.2 this interpretation is revisited in light of
additional evidence suggesting other mechanisms may play a dominant role.
Lastly, the underestimation of the subtle anti-clockwise turning of the wake depends on turbulent mixing (Fig. 6b). For the
wake turning, this behavior results from downward turbulent momentum entrainment, which transports more veered wind from
aloft into the wake (van der Laan and Sørensen, 2017; Gadde and Stevens, 2019; Englberger et al., 2020; Stieren et al., 2022;
Kasper et al., 2024). Accordingly, cases with weaker entrainment, such as TKE000 and BASE, show a stronger directional bias.
Although the absolute differences in wind direction are small (about $1°$ between TKE000 and TKE100), over long distances
these differences may accumulate into downstream power losses for adjacent wind farms.

### 3.5   Under-resolved spatial gradients in the NWP-WFP simulation wake

In this section, we examine spatial gradients in the wind farm wake as represented by the LES at full resolution and when
spatially averaged to match the grid resolution of the BASE case. We also compare the LES results with the NWP-WFP cases
BASE and TKE100. The goal is to evaluate how the realistic spatial gradients of the LES influence wake recovery, and how
the horizontal and vertical gradients of the latter compare with the NWP-WFP simulations.
The difference between the coarsened LES and the BASE and TKE100 profiles of wind speed grows in the streamwise
distance. Figure 8 shows hub-height profiles of wind speed (Figs. 8a–e) and TKE (Figs. 8f–j) along the spanwise ($Y$) direction
at selected streamwise distances: the wind farm entrance ($X = 0$ km), middle ($X = 5$ km), near exit ($X = 10$ km), and two
downstream locations ($X = 15$ and $20$ km). At the entrance ($X = 0$ km), the spatially averaged LES gradients in wind speed
(coarsened LES) agree reasonably well with both NWP-WFP cases (Figs. 8a). However, deviations become noticeable at
$X = 5$ km (Figs. 8b), especially for the BASE case, and grow progressively larger through $X = 10$ and $15$ km (Figs. 8c and d).
At $X = 20$ km (Figs. 8e), the discrepancy remains large. The narrow band of reduced wind speed and increased TKE near the
center of the wind farm in the $Y$ direction is created by the existence of two turbines within that grid cell (Fig. 2b), as discussed
in Section 3.3.
From the standpoint of spatial wind speed gradients, differences between the LES and the NWP-WFP cases are striking.
While the coarsened LES profile appears similar to those of the NWP-WFP cases within the wind farm ($0 \leq X \leq 10$ km,
$0 \leq X/D \leq 56$), this resemblance is misleading and fails to represent important physical processes. The full-resolution LES
wind speed profiles feature sharp gradients that resemble spikes (Figs.8a–c). These strong local gradients facilitate faster wake
recovery in the LES. Notably, model discrepancies in wind speed profiles grow with streamwise distance while these spiky
gradients persist ($0 \leq X \leq 10$ km, $0 \leq X/D \leq 56$), but no longer increase between $X = 15$ and $20$ km ($X/D = 84$ and $112$,
respectively), where the LES gradients become smoother.
This behavior suggests that the fundamental problem lies upstream, within the intra-farm and near-farm wake regions. Once
the spikiness disappears in the LES, the differences between models stabilize. Therefore, the persistent lower wind speeds in
NWP-WFP simulations in the far wake of the farm (Figs. 8d,e) are not caused by conditions there, but rather reflect limitations
in representing spatial gradients and turbulent mixing within the intra-farm and near-farm wake regions. As conceptually

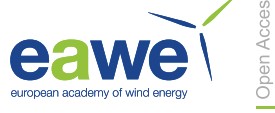

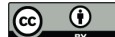

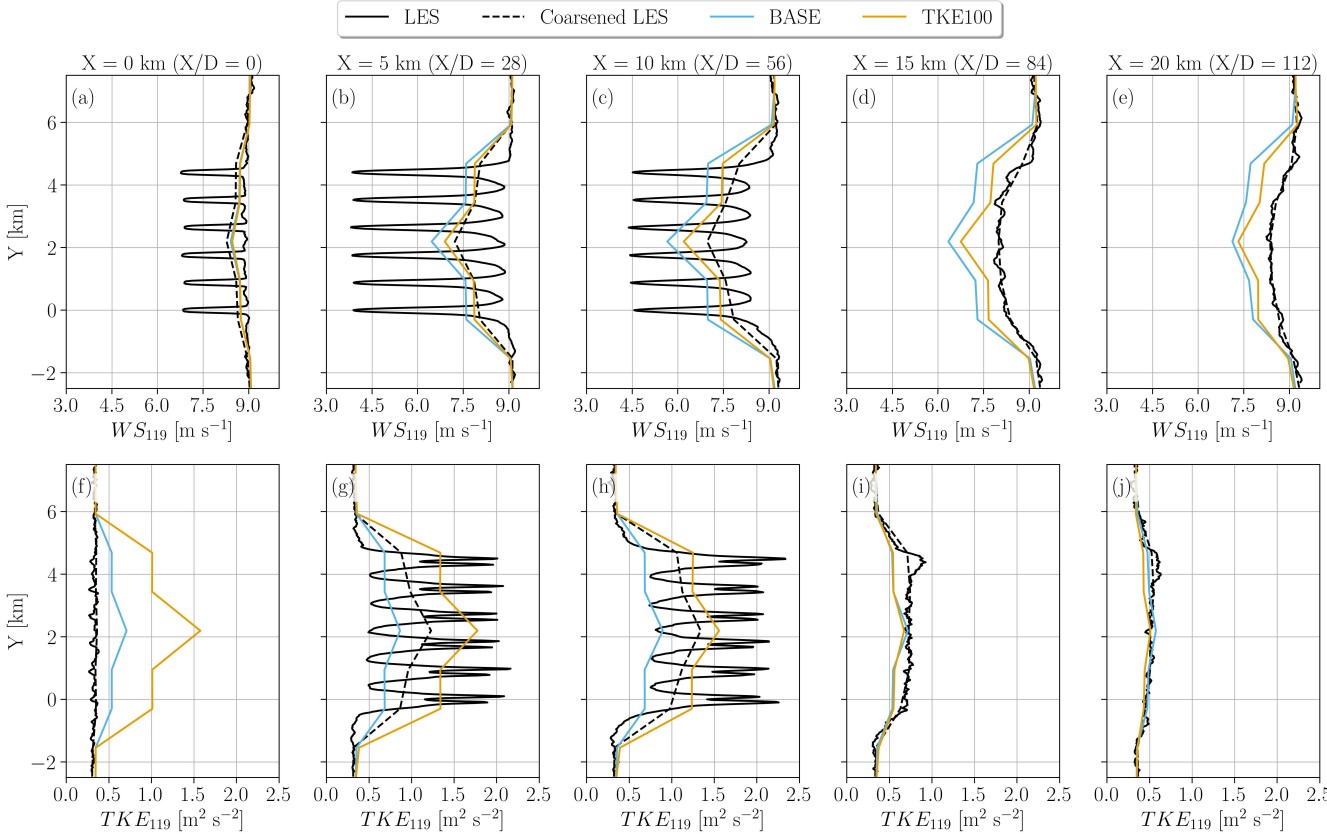

**Figure 8.** Time-averaged wind speed (a–e) and TKE (f–j) at hub height (119 m AGL) along spanwise $Y$ lines at specific streamwise distances $X$ (expressed as $X/D$ in parentheses) for the LES, coarsened LES, and NWP-WFP BASE and TKE100 cases.

illustrated in the Introduction, the slow wake recovery (Figs. 1b,c) and rapid TKE decay (Figs. 1e,f) are evident in the results shown in Figs. 8c,d and h,i, respectively.

Vertical profiles reveal how wind farm effects modify wind speed (Figs. 9a–f) and TKE (Figs. 9g–l) gradients in each simulation. Vertical profiles are sampled along the southernmost turbine column (near $Y = 0$ km) at selected streamwise locations to examine how each simulation resolves vertical gradients. The data are not spatially averaged, allowing direct assessment of the gradients resolved by each grid. Upstream of the farm, wind speed (Fig. 9a) and TKE (Fig. 9g) profiles are nearly identical across simulations. Within and downstream of the farm, momentum extraction (Figs. 9b–d) and TKE production (Figs. 9h–j) alter these profiles. In the LES, the stronger wind speed deficit enhances vertical shear, especially near the rotor top tip, by $X/D = 28$ (Figs. 9b–d), leading to intense TKE generation in the same region (Figs. 9i–j). Combined, the stronger vertical wind speed gradient and enhanced TKE in the LES contribute to a faster intra-farm and near-farm wake recovery.



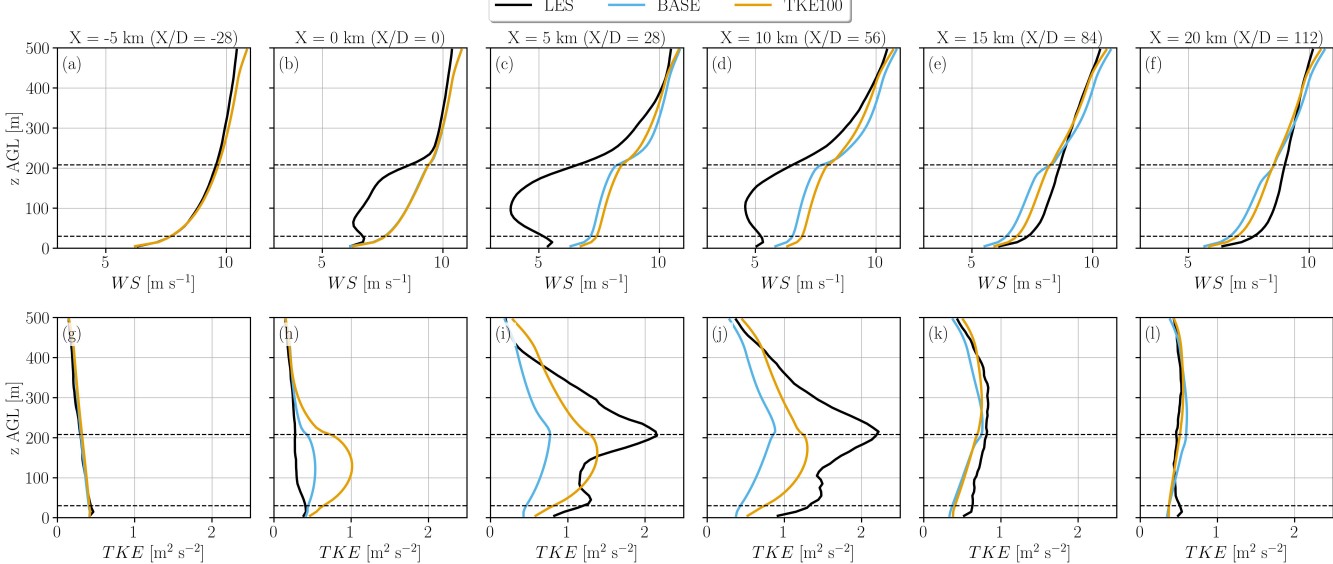

**Figure 9.** Vertical profiles of time-averaged wind speed (a–f) and TKE (g–l) at specific streamwise distances $X$ (expressed as $X/D$ in parentheses) probed over the southernmost column of turbines near $Y = 0$ km. The LES at full resolution and NWP-WFP cases BASE and TKE100 are shown. The dashed horizontal lines represent the rotor top and bottom tips.

Despite the fine vertical resolution, NWP-WFP simulations fail to capture strong vertical gradients due to coarse horizontal
resolution. Although the NWP-WFP simulations use the same vertical resolution as the LES ($\Delta z \sim 10$ m), they locally under-
resolve vertical wind speed gradients (Figs. 9b–d). The LES shows a much sharper local wind speed deficit (Figs. 9b–d), while
in the NWP-WFP simulations the deficit is diluted due to spatial averaging over the coarser horizontal grid ($\Delta X \sim 1$ km).
This horizontal averaging dilutes the momentum sink, weakening both horizontal and vertical gradients. As a result, even with
sufficient vertical resolution, the vertical structure of the wake is poorly captured in the NWP-WFP simulations due to the
coupling between horizontal and vertical gradients.
Further evidence of the critical role of under-resolved gradients in intra-farm and near-farm wake recovery emerges when
evaluating the effect of TKE. From a turbulent mixing perspective, increased TKE accelerates wake recovery: for instance,
the TKE100 case displays faster recovery than the BASE case, as also shown by Rosencrans et al. (2024). However, despite
exhibiting more TKE than the coarsened LES throughout much of the wind farm ($0 \leq X \leq 10$ km), including earlier onset
of added TKE (Fig. 8f), the TKE100 wake still recovers slower than that of the LES. These findings highlight that enhanced
farm-added TKE alone is insufficient to compensate for the under-resolved gradients.



# 4 Discussion

## 4.1 What are the fundamental problems?

The preceding results identify two interconnected sources of modeling bias: (i) under-resolved spatial gradients in turbine wakes and (ii) rapid decay of farm-added TKE in NWP-WFP simulations. *Under-resolved*, not *unresolved*, is the appropriate term, since the mesoscale grid partially captures the turbine-induced wind speed gradients, but lacks sufficient fidelity. These two issues affect the physics of wake recovery, which is governed by two interrelated mechanisms:

(a) the magnitude of turbulent mixing, and

(b) the strength of spatial gradients in the mean wind velocity field.

The recovery of momentum in wind turbine and near-farm wakes occurs through lateral and vertical turbulent entrainment of momentum, expressed as divergences of $\overline{u'v'}$ and $\overline{u'w'}$, respectively (Vermeer et al., 2003; Cal et al., 2010; Calaf et al., 2010; Abkar and Porté-Agel, 2015; Porté-Agel et al., 2020; Stieren and Stevens, 2022; van der Laan et al., 2023). Here, $u'$, $v'$, and $w'$ denote fluctuations relative to the time-averaged wind velocity components $U$, $V$, and $W$ in the streamwise, spanwise, and vertical directions, respectively. These momentum fluxes are commonly approximated using the Boussinesq hypothesis (Boussinesq, 1897; van der Laan et al., 2023) or equivalently the K-theory (Stull, 1988):

$$\overline{u'v'} = K_m \frac{\partial U}{\partial y} \tag{1}$$

$$\overline{u'w'} = K_m \frac{\partial U}{\partial z}. \tag{2}$$

Here, $K_m$ is the eddy viscosity, which depends on turbulence and atmospheric stability (Stull, 1988), and $\frac{\partial U}{\partial y}$ and $\frac{\partial U}{\partial z}$ are the spanwise and vertical gradients of wind speed (for convenience, we assume the mean flow aligns with the $x$-direction). The eddy viscosity $K_m$ increases with TKE, and therefore wakes tend to recover more rapidly under convective conditions compared to neutral or stable conditions (Magnusson and Smedman, 1994; Iungo et al., 2013; Fitch et al., 2013a; Mirocha et al., 2015; Abkar and Porté-Agel, 2015; Stevens and Meneveau, 2017; Bodini et al., 2017; Lundquist et al., 2019; Rosencrans et al., 2024; García-Santiago et al., 2024). The two modeling biases identified above directly impact these physical mechanisms of wake recovery:

(i) Under-resolved spatial gradients: weaken the mean shear in the wake, thereby reducing the turbulent fluxes [affecting factor (b)]. Even if TKE is present, the lack of strong horizontal and vertical gradients limits momentum transport into the wake.

(ii) Rapid decay of farm-added TKE: reduces the eddy viscosity $K_m$, which diminishes the efficiency of turbulent mixing [affecting factor (a)]. As a result, the momentum recovery by turbulence is suppressed, even in regions where spatial gradients are better resolved.



In the following subsections, we examine each of these two problems in more detail. We begin with the role of under-
resolved wind speed gradients and their consequences for wake recovery, followed by the impact of rapid TKE decay in the
farm region.

### 4.1.1   Under-resolved wind speed gradients

The first and perhaps most critical problem is that NWP-WFP simulations cannot accurately represent the spatial wind speed
gradients that drive intra-farm and near-farm wake recovery. Many studies have shown that WFPs can reproduce wind speed
deficits at turbine grid cells that generally resemble those from LES (Vanderwende et al., 2016; Abkar and Porté-Agel, 2015;
Archer et al., 2020; Peña et al., 2022; García-Santiago et al., 2024). However, a closer look at the downstream regions reveals
a slower wake recovery in NWP-WFP simulations compared to LES (e.g., Figs. 4a,c in Vanderwende et al. (2016); Fig. 7 in
Archer et al. (2020); Figs. 11–13 in Peña et al. (2022); Figs. 4 and 7 in García-Santiago et al. (2024)). Here, we demonstrate that
this discrepancy arises because the momentum sink term in WFPs is not resolved by the grid, and thus is not severely affected
by its coarseness, whereas the wake recovery is fundamentally resolved by the grid. Specifically, the NWP-WFP simulations
exhibit inherently weaker horizontal (Figs. 8a–c) and vertical (Figs. 9b–d) gradients in the wind velocity field within the intra-
farm and near-farm wake regions, i.e. $\frac{\partial U}{\partial y}|_{\text{NWP}-\text{WFP}} << \frac{\partial U}{\partial y}|_{\text{LES}}$ and $\frac{\partial U}{\partial z}|_{\text{NWP}-\text{WFP}} << \frac{\partial U}{\partial z}|_{\text{LES}}$, respectively. As a result,
even when farm-added TKE levels are comparable to or exceed those in the LES, the intra-farm and near-farm wake recovery
(driven by the weaker gradients) remains underestimated in the NWP-WFP simulations (Sections 3.4 and 3.5). The fact that
simulations with the finest grid resolution (DX2D and DX2DTKE100), which better resolve spatial gradients, improve the
representation of the near-farm wake recovery (Figs. 7a,d) supports this argument. Under-resolved gradients have also been
noticed by others (Fischereit et al., 2022b). Related, in another study using WRF and the Fitch WFP (Pryor et al., 2020),
shorter wind farm wakes result from finer grid resolutions, which could in part be explained by the impact of the under-resolved
gradients on wake recovery.

### 4.1.2   Rapid decay of farm-added TKE

The second problem is the relatively fast decay of farm-added TKE in the NWP-WFP simulations compared to the LES. This
rapid decay reduces turbulent mixing, effectively lowering the eddy viscosity $K_m$, and further slowing wake recovery. Even
when large amounts of turbine-added TKE are injected (e.g., $\alpha = 75\%$ or $100\%$), the TKE fails to accumulate row-by-row
as it does in the LES (Fig. 6c). Across all tested TKE enhancement levels ($\alpha = 25\%$–$100\%$), the NWP-WFP TKE decays to
near-ambient levels within 5 km downstream, almost twice as fast as in the LES.
A compelling hypothesis is that this rapid decay is not merely a dissipation artifact but rather a consequence of the first
problem – the under-resolved gradients. The first evidence for this hypothesis is the excellent agreement between NWP-WFP
and LES TKE levels in the inflow profiles (Fig. 3d) and in the far wake (Fig. 6c), indicating that the mesoscale model does
not inherently over-dissipate TKE. Instead, the issue may lie in the intra- and near-farm wake regions, where TKE fails to
persist because the shear production of TKE depends on spatial gradients (Mellor and Yamada, 1982; Nakanishi and Niino,
2009), which are under-resolved. For instance, the TKE levels are enhanced in regions of strong wind shear due to shear





production of TKE. If the wind speed and direction became homogeneous (zero wind shear), the TKE profile as in Fig. 3d would decay. Therefore, an explicit source of TKE without sufficient gradients to maintain it becomes unsustainable and decays (Fig. 5l). This distinction highlights a key limitation: while the NWP-WFP approach can represent background ABL turbulence reasonably well, it fails to represent the strong turbulence generated within the wind farm itself. The mechanism for the rapid decay of TKE is relevant for any studies that employ the NWP-WFP approach, especially those that propose modifications to the turbine-added TKE source (Fitch et al., 2012; Archer et al., 2020; Khanjari et al., 2025).

## 4.2 Implications

Because both problems – slow wake recovery and fast TKE decay – are fundamentally linked to grid resolution, they are likely to affect wake recovery not only in WRF with the Fitch WFP but also in other NWP and climate models that use WFPs. As such, these findings underscore the need for improved representation of both spatial gradients and turbulent mixing if WFPs are to accurately simulate wind farm wakes and their downstream impacts. Building on this, we present implications of our work for studies of real wind farm wake effects, where environmental and observational aspects increase the complexity of the analysis. We highlight the importance of separating wake generation (momentum extraction) from its recovery when evaluating model performance.

Some NWP-WFP simulations driven by realistic synoptic conditions have demonstrated reasonable agreement with observed wind speed deficits in wind farm wakes, based on aircraft measurements (Siedersleben et al., 2018a, b, 2020; Larsén and Fischereit, 2021; Ali et al., 2023) and SCADA data (Sanchez Gomez et al., 2024) from offshore sites. However, discrepancies in inflow conditions between simulations and observations remain a major challenge, as they can obscure the evaluation of wake recovery. Despite this, several case studies have achieved good correspondence with observed wind speed profiles along wind farm transects. Given the inherent spatio-temporal variability and limited control over inflow in realistic configurations, such results are promising. In contrast, our idealized setup ensures excellent agreement in the inflow, allowing a more direct assessment of WFP performance and limitations. These controlled conditions provide valuable guidance for improving their representation in more complex, operational scenarios.

Another important issue is the separation between wake generation and recovery for modeling evaluation. The wind speed bias between the NWP-WFP simulation and the LES within the wind farm in not exclusively caused by a difference in wake recovery, but also by a difference in momentum extraction. For instance, in Fig. 4b, the coarser-resolution cases BASE and DX5D have larger row-averaged power (extract more momentum) than the finer-resolution cases DX2D and DX2DTKE100. As a result, in combination with differences in wake recovery between the simulations, cases BASE and DX5D have stronger wind speed deficits (Fig. 7a). Matching the wind speed deficit predicted by LES or observations does not necessarily mean the dynamics of wake recovery are well represented in NWP-WFP simulations. For instance, a seemingly faster wake recovery was attributed to the NWP-WFP simulation in comparison with the LES, when in fact the NWP-WFP simulation generated a much weaker wake in the first place (Eriksson et al., 2015). As the stronger wake of the LES recovers momentum, it eventually matches the wind speed deficit of the weaker wake of the NWP-WFP simulation downstream. Thus, wake generation and recovery are two important aspects of wind farm flows that need to be considered simultaneously.





The degree of wake recovery underestimation is likely sensitive to inflow wind speed, direction and atmospheric stability.
For instance, the weaker ambient turbulence in stable conditions promotes longer individual turbine wakes (and thus stronger
spatial gradients in wind speed) in comparison with neutral or convective conditions (Abkar and Porté-Agel, 2015; Abkar et al.,
2016b). Hypothetically, these unmixed wind speed gradients could further slow wake recovery in NWP-WFP simulations,
which requires more research. Evaluating wake recovery in stable conditions is important because it is exactly when wakes are
most pronounced (Magnusson and Smedman, 1994; Iungo et al., 2013; Fitch et al., 2013a; Mirocha et al., 2015; Abkar and
Porté-Agel, 2015; Bodini et al., 2017; Lundquist et al., 2019; Rosencrans et al., 2024; García-Santiago et al., 2024).

## 5   Conclusions

Numerical Weather Prediction (NWP) and climate models equipped with Wind Farm Parameterizations (WFPs) provide a
powerful framework for simulating wind farm cluster wakes, both onshore and offshore. Their ability to capture physical
processes across a wide range of spatiotemporal scales enables studies of atmosphere–wind farm interactions under realistic
conditions and over large domains. In this paper, we assess the strengths and limitations of the NWP-WFP approach using
the Weather Research and Forecasting (WRF) model with the Fitch WFP, in comparison to the neutrally-stratified large-eddy
simulation (LES) of Kasper et al. (2024). We find that near-farm wake recovery is slower in the NWP-WFP simulations relative
to the LES for two interconnected reasons: (i) under-resolved spatial gradients in the wind velocity field and (ii) overly rapid
decay of wind turbine-added turbulence kinetic energy (TKE).
The first issue stems from the fact that in the LES, near-farm wake recovery is driven by sharp velocity gradients and
enhanced turbulent mixing that replenishes momentum in the wake. In contrast, while the NWP-WFP momentum deficit at
the turbine grid cell is *unresolved* by the grid, its downstream recovery depends on grid-resolved gradients. However, these
horizontal and vertical gradients in wind speed in the intra- and near-farm wake are *under-resolved* at the mesoscale grid
spacing used in NWP-WFP, leading to underestimated wake recovery. In coarser-resolution simulations, the spanwise-averaged
wind speed bias in the near-farm wake reaches approximately $-1.0$ m s$^{-1}$, and about $-0.6$ m s$^{-1}$ in the far wake, relative to
the LES. Finer-resolution cases reduce these biases to around $-0.3$ to $-0.5$ m s$^{-1}$ in the near-farm wake and approximately
$-0.4$ m s$^{-1}$ in the far wake. Despite similarly fine vertical grid spacing between the NWP-WFP and LES, horizontal smearing
of the momentum deficit in the NWP-WFP also results in underestimated vertical wind speed gradients. Even simulations with
increased turbine-added TKE could not fully compensate for these under-resolved gradients. Thus, under-resolved gradients in
both horizontal and vertical directions limit wake recovery.
The second issue involves the decay of wind turbine-added TKE. In the LES, TKE accumulates progressively along the wind
farm rows, peaking near the wind farm exit. In the NWP-WFP simulations, however, the maximum occurs within the first few
rows and then decreases monotonically along the streamwise direction because the TKE cannot be maintained. This rapid TKE
decay continues in the near-farm wake region (within 5 km downstream), likely due to the same under-resolved gradients.
The inflow profiles have similar shear and veer characteristics across simulations, which initially produce comparable TKE
levels. However, the insufficient magnitude of spatial gradients in the intra- and near-farm wake in the NWP-WFP renders the





turbine-added TKE unsustainable. As a result, the TKE decays much faster than in the LES and undermines wake recovery in
these neutrally-stratified simulations.
A key insight is that the slow recovery in the near-farm wake, although confined to a short downstream distance, has lasting
consequences for the far wake region. In coarse-resolution simulations, the far wake wind speed bias remains substantial even
when large amounts of turbine-added TKE are introduced. However, when near-farm wake recovery improves through finer
grid resolution and better-resolved gradients, the far wake bias is also reduced. This highlights the importance of accurately
representing wake recovery not only near the farm, but also as a prerequisite for improving conditions further downstream.
Because these two findings stem from the relatively coarse mesoscale grid, they are relevant to other NWP and climate
modeling frameworks that employ any form of WFP. Beyond improving the subgrid parameterizations of momentum sinks
and TKE sources, it is equally important to develop new strategies that account for the effects of under-resolved gradients in
NWP-WFP frameworks. These efforts should produce spatial variations of TKE and wake recovery rates more consistent with
LES. Otherwise, the underestimation of wake recovery may lead to overestimated wake losses.
Future work could: (i) propose methods to address the problem of under-resolved gradients, (ii) investigate the differences
between NWP-WFP and LES under aligned and staggered turbine layouts, and (iii) assess wake recovery under varying at-
mospheric stability regimes. Another possible direction is to (iv) investigate whether different planetary boundary layer (PBL)
schemes can improve turbulent mixing and thus wake recovery, such as the 3DPBL scheme (Kosović et al., 2020; Juliano et al.,
2022). Regarding (iii), we note that the velocity gradients in the LES become smoother and more mesoscale-resolvable beyond
approximately 5 km downstream. Under such conditions, NWP-WFP simulations may better reproduce the reference LES
recovery. Following this rationale, convective boundary layers, by mixing sharp gradients more efficiently, may allow NWP-
WFP to more accurately capture wake recovery than is possible under neutral or stable conditions. Therefore, understanding
the role of atmospheric stability on wake recovery is also necessary for improving the accuracy of NWP-WFP approaches.
*Code and data availability.* The WRF model with the axial induction correction (Vollmer et al., 2024) version 4.4 is available at https:
//github.com/wradunz/WRFv4.4-DRM_GAD.git. The WRF setup files for the NWP-WFP simulations can be accessed at Radünz (2025).
**Appendix A:  DTU 10-MW turbine power and thrust curves**
This section evaluates the sensitivity of wake recovery to the thrust coefficient ($C_T$), as outlined in Table 1. The power and
thrust coefficient curves for the DTU 10-MW wind turbine are shown in Figs. A1a,b, respectively. For an inflow wind speed of
approximately 9 m s$^{-1}$, the turbine produces slightly over 5 MW of power, and the corresponding $C_T$ is approximately 0.81.
The value of $C_T = 0.75$ adopted in the LES (Section 2.1) yields wind speed deficits similar to those obtained with a fixed
$C_T$ of 0.81 or with a wind-speed-dependent $C_T$ (Fig. A2a). In contrast, using a lower $C_T$ value of 0.50 results in a weaker
wake and reduced TKE generation (Fig. A2c), since the TKE source term is proportional to the difference $C_T - C_P$. Therefore,





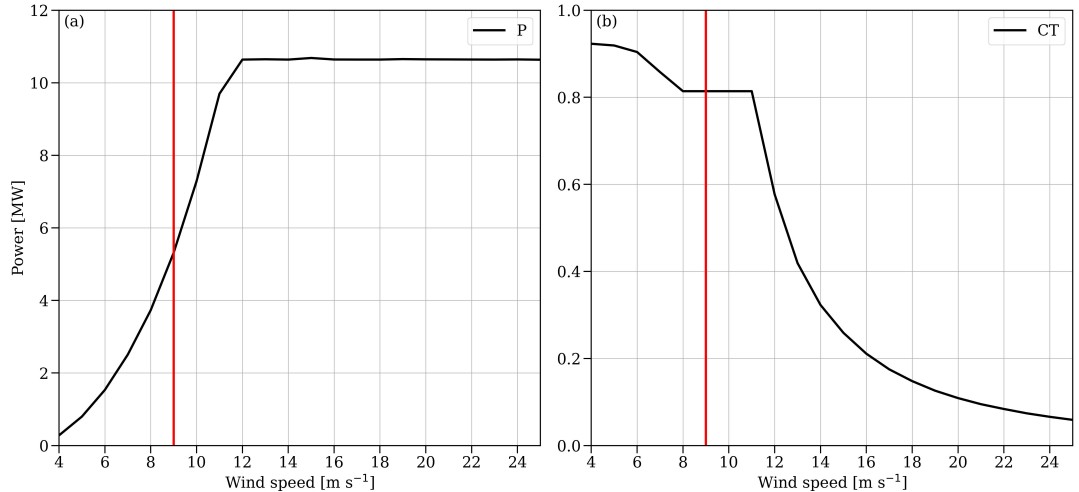

**Figure A1.** Power (a) and thrust coefficient (b) curves as function of wind speed for the wind turbine of the DTU 10-MW model. The red vertical line denotes the point of operation based on the hub height wind speed of approximately 9 m s$^{-1}$ considered in this study.

whether $C_T$ is set to 0.81, 0.75, or dynamically determined based on wind speed, the main conclusions of our investigation
remain unchanged.
*Author contributions.* Conceptualization: WCR and JKL. Methodology: WCR. Data curation: WCR and JHK. Formal analysis: WCR. In-
vestigation: WCR and JKL. Writing (original draft): WCR and JKL. Writing (review and editing): all authors. All authors contributed to the
discussion and interpretation of results.
*Competing interests.* At least one of the (co-)authors is a member of the editorial board of *Wind Energy Science*.
*Acknowledgements.* This work was partially supported by the U.S. Department of Energy, Office of Science Energy Earthshot Initiative, as
part of the Addressing Challenges in Energy—Floating Wind in a Changing Climate (ACE-FWICC) Energy Earthshot Research Center. A
portion of the research (WRF simulations) was performed using computational resources sponsored by the Department of Energy's Office
of Energy Efficiency and Renewable Energy and located at the National Renewable Energy Laboratory. This work was authored in part by
the National Renewable Energy Laboratory operated for the U.S. Department of Energy (DOE) under Contract No. DE-AC36-08GO28308.
This project has received funding from the European Research Council Horizon Europe program (Grant No. 101124815).





**Figure A2.** Streamwise variation of spanwise averages over the wind farm area of hub-height wind speed (a), direction (b), TKE (c), and streamwise gradient of wind speed (d) for the LES at full resolution, coarsened LES, and NWP-WFP cases with different $C_T$. In the CTTC case, the thrust coefficient $C_T$ varies with wind speed according to the turbine's thrust curve. The colored background areas indicate the streamwise extent of the intra-farm wake (gray), near-farm wake (green), and far wake of the farm (blue).

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
