# Peer review of "Under-resolved gradients: slow wake recovery and fast turbulence decay with mesoscale Wind Farm Parameterizations"

_Wind Energy Science, 2025_

## Referee Comment (RC1)

The manuscript "Under-resolved gradients: slow wake recovery and fast turbulence decay with mesoscale Wind Farm Parameterizations" by Radünz et al. compares LES of a regular-spaced idealised offshore wind farm in a conventially neutral boundary layer with the same wind farm under the same meteorological conditions in an idealised setup of the mesoscale model WRF.

In the mesoscale model different grid resolutions and parameters of the wind farm parametrization are studied. The authors compare these different setups to the LES results and conclude that the wake dissipates slower in the mesoscale model because of under-resolved vertical gradients of wind speed and a decay of TKE that is occurring in a smaller distance downstream of the wind farm compared to the LES.

The authors address a relevant topic as mesoscale modelling has become one of the cornerstones of studying the effect of expanding wind energy employment on power output and the state of the atmosphere. The manuscript is well written, and the different steps are easy to understand.

However, I do not think the authors provide enough evidence for their strong conclusion that the mesoscale model is under-representing the wake decay. The reason for my doubt lies in a fundamental flaw in the study setup.

Figure 4a clearly shows the difference in power production of the wind turbines in the LES and the mesoscale model. The difference in power production (in the BASE case it is about 20% more power) results in a significant difference in removed momentum between the models. I find it hard to be convinced that the large difference of wake deficit especially visible in Figures 6a and 8e is not mainly due to this difference in removed momentum. I would actually claim that Figure 6d reveals that the wake decay is quite comparable outside of one grid point just downstream of the wind farm.

Furthermore, the study is based on a very specific case of wind farm flow. The situation that all turbines are aligned in quite close distance to each other and experience full wake exposure of all upstream turbines is in reality very rare. Much more often the wind turbines are actually operating in partial wake or in the wake of a much more distant upstream turbine. In addition, wind farms with regular layouts like the studied one are rarely to be found. For this particular case the mesoscale models stands no chance to replicate the power production of the LES, as the momentum extracted of each turbine is equally distributed over the whole cross-section of the grid cell. Thus, I would not recommend to base general conclusions of the mesoscale model's ability to model wind farm flow on this particular extreme flow case.

Further comments:

- Title: "Under-resolved gradients: slow wake recovery and fast turbulence decay with mesoscale Wind Farm Parametrizations" - I strongly oppose the idea that the wind farm parametrization should be responsible for the wake recovery and the calculation of turbulence downstream of the wind farm. This job should be solely on the PBL scheme. Thus, I find the title misleading.
- Description of the PBL scheme: A description of how the PBL scheme uses TKE for calculating mean momentum transport as well as a description how the PBL scheme itself is calculating TKE is missing. I think this is crucial to get an idea about what is happening in the mesoscale model. In fact, the used PBL-scheme is not even mentioned in the manuscript.
- L. 65 ff. – I don't like this part of the introduction and Fig.1 – I think this should be entirely moved to the discussion and conclusion. It is making assumptions about the wake

recovery that are actually part of the study. As I mentioned earlier, I do not see enough evidence of a consistent underestimation of wake recovery of wind farms by the WRF-WFP. In fact, comparison with power data (Sanchez-Gomez et al. 2024) show a quite good performance.

- L.79: "overestimation of power losses often reported when using WFPs" – Montavon et al. 2024 relate the overestimation of power losses mainly to the error in calculating gross yield which was addressed with the correction in Vollmer et al., 2024. In fact, they find the patterns of external as well as internal wakes quite well replicated by the WRF WFP. In general I would recommend to treat any reported differences in energy yield before this correction carefully and not relate them to higher wake losses.

- L.231 ff: The reason for the different development of power production along the turbine rows needs to be explained more precisely. The consequences for drawing any conclusions about wake recovery need to be made aware of.

- Figure 5i – This figure clearly shows one of the weaknesses of the MYNN-PBL scheme at high grid resolution – there is no horizontal diffusion of momentum, so the individual turbine wakes persist throughout the whole domain. I think the manuscript should focus much more on this limitation. Also, I do not understand why the control volume has to be smaller for this particular case. I would argue it should be the same as in the other cases to allow for a fair comparison.

- Figures 6a & 7a – In the LES the wind recovers by the end of the domain, even reaching a higher wind speed than upstream of the farm. In the mesoscale model the wind never recovers to upstream conditions. Can any influences of the boundary conditions in the models be excluded as a reason for this discrepancy?

- L.301 ff: In the end, there is a pronounced difference of wake recovery at exactly one grid cell downwind of the farm (Fig. 6d). It would be interesting to understand what this "key mechanism" is.

- Figure 9: I think the comparison along one column of turbines is misleading. If a location exactly between the turbine rows would have been selected, the TKE in the mesoscale model would actually be higher than in the LES, as indicated by Fig. 8g. The meaningful comparison in my opinion would be the cross-stream averaged profiles.

- L. 510 ff. – I am not convinced that the difference occurring at exactly one grid cell is responsible for much of the observed differences in the wake deficit everywhere else.

The study has some interesting data to work with, that has the potential to identify key limitations in the PBL-scheme of the mesoscale model to model a wake-recovery comparable to higher fidelity models. I disagree with the conclusion of the manuscript, that a general slower wake recovery in the mesoscale model is and can be identified with these data. If the manuscript clearly addresses the key limitations of the comparability of the two data sets and focuses on the wake recovery processes related to the PBL scheme, it has the potential to add a valuable contribution to the discourse.

---

## Referee Comment (RC2)

**Review of WES-2025-147: "Under-resolved gradients: slow wake recovery and fast turbulence decay with mesoscale Wind Farm Parameterizations"**

This was a difficult review. There are many good contributions in this work: the manuscript is exceptionally well written, the approach is logical, and the figures are really effective. Many simulations had to be conducted, stored, post-processed, and analyzed for this work and I am painfully aware of what that means. But at the end I cannot help but concluding that there is very little new information in it and, actually, quite a few attempts to "sell" new concepts that are really not new and, if anything, misrepresent the real problems.

- 1. The title is misleading: by using the plural ("parameterizations") one is led to believe that multiple WFPs are going to be evaluated in the study, whereas only one, that by Fitch et al. (2012), has been assessed. There is no explanation as to why only Fitch's was chosen or why the other three WFPs included in WRF (MAV parameterizations) were not considered. Only the sensitivity to the correction factor  $\alpha$  for TKE was evaluated, but that does not change the WFP, still Fitch's. Based on this and the other issues below, I have concluded that this paper is a sensitivity study of the Fitch parameterization to one tuning parameter and cannot be generalized to any other WFP.
- 2. The paper proposes some sort of a new interpretation of the malfunctioning of the Fitch parameterization: the authors name it "under-resolved gradients". They basically claim that these under-resolved gradients are the fundamental reason why WFPs (but in reality it's just Fitch's) do not resolve the wind speed deficit and added TKE patterns accurately. This is not correct. One could simply replace the term "under-resolved gradients" with "sub-grid wakes" and none of the proposed findings would be new anymore. It is not the local gradients per se that cause differences in the wake recovery, it is the missing wakes; the local gradients are the obvious and inevitable consequence of having a wake with a localized wind speed deficit. There would be no gradient if there was no wake, obviously. By shifting the attention to the gradients, the true issue disappears, which is that the wake effects need to be parameterized better than how it's done in Fitch's. The wakes are missing, therefore (obviously) the gradients are missing, so let's focus on the missing wakes, not on the missing gradients.

Why are the gradients missing or under-resolved? Because, with the Fitch WFP, the wind speed deficit added by the turbines is smeared in the volume of the grid cell and therefore it is not possible to resolve or create the proper y- or z-gradients. This is an implicit limitation that is obvious and is not the cause, but rather the direct consequence, of the WFP.

Furthermore about the gradients: the paper describes them as some sort of an LES feature that is almost undesirable ("the LES ... wind speed profiles feature sharp gradients that resemble spikes" or "Once the spikiness disappears, the differences between the [LES and WRF] models stabilizes"). The LES do not simulate just the

gradients, they simulate the full wake and therefore gradients appear. The gradients are the effect, not the cause.

Lastly about the gradients: as described in the manuscript, these gradients are a function of  $\Delta y$  (and  $\Delta z$ ), thus a direct comparison between LES and WRF is not correct because the two models have different resolutions and different  $\Delta ys$ . But one can look at the magnitude of gradients in the Coarsened LES and compare them to those in WRF. In the figure below, which is a zoomed version of Fig. 5, one can notice that the gradients in WRF on the right (U@1 – U@3)/ $\Delta y$  or (U@2 – U@1)/ $\Delta y$  are actually larger than those in the LES (on the left). The whole discussion in Section 4.1.1 is therefore moot.

- 3. The second focus of the paper is the so-called "rapid decay" of TKE in the resolved wake in WRF with Fitch's. This terms is improper because it implicitly assumes that it is sufficient to add TKE in the grid cells of the wind turbines and then the resolved processes will just advect and redistribute this TKE downwind and, as such, the issue is just that this happens too quickly. Basically, the (wrong) assumption here is that the source of TKE is at the turbine. While this is absolutely true for the wind speed deficit and partially true for a single turbine's TKE in the near-wake via tip vortices, it is absolutely not true in the far wake. The authors are in fact aligned with the literature when they recognize the well-known fact that (l. 438-439) "the shear production of TKE depends on spatial gradients, which are under-resolved" but they are incorrect in stating that "TKE fails to persist" there. Instead, it is not FORMED there because of the under-resolved shear. It's not a decay problem, it's a missing addition problem.
- 4. Ultimately the true issue is: what are the authors proposing to resolve these issues? How can we resolve these gradients better? After all these computationally expensive

and time consuming simulations to, in my opinion, demonstrate the obvious, what is the solution to the issue? None is proposed. I do not see the ultimate value of this paper to the scientific community, it fails to show novelty except semantically, and it does not provide guidance, let alone a solution, on how to mitigate the issue.

5. As a final note, I find it extremely inaccurate to state that: "Beyond improving the subgrid parameterizations of momentum sinks and TKE sources, it is equally important to develop new strategies that account for the effects of under-resolved gradients in NWP-WFP frameworks." Again, the gradients are under-resolved precisely because the wakes are not well parameterized. These are not two separate issues, they are the same issue. By focusing the attention on one of the effects of the problem, i.e., the gradients, rather than the root cause, i.e., the misrepresented wakes, this paper ultimately has no use and therefore should not be published.

**Minor issue**

Figure 5: there might be some errors here. First of all, why is TKE zero in the black column right upstream of the wind farm in panel (h)? For TKE to be zero, added TKE would need to be negative, is this even possible? In panel (l), I would expect a very small error in the last column of the wind farm in the center row where the BASE case shows a grid cell with the highest TKE with the same yellow shade as the Coarsened LES in (d). Why is it in the darkest blue instead, indicating the largest error? The errors should also be smallest along the center row, but there is no such feature.

---

## Referee Comment (RC3)

**Review comments**

The paper is generally well written and structured. The figures, results, and comparisons between simulations and reference data are strong. I compliment the authors for the framework and the simulation effort. However, the title, the claims, and the arguments in the discussion and conclusions are not sufficiently supported by the presented results.

If the study were reframed to focus on a more specific aspect—such as "Effect of the added TKE on wake recovery in the Fitch parameterization" (to give an example)—I will immediately recommend acceptance. In its current form, the manuscript does not fully reflect what is stated in the title and abstract. Substantial content changes are needed to align objectives, title, and conclusions.

Below is a list of major and specific comments:

**Major comments**

- 1. Throughout the paper, mesoscale resolution is presented as the main cause of what is referred to as "under-resolved" gradients. This interpretation is problematic because coarse resolution effects are expected—you can not resolve something that is smaller than your grid. Mesoscale simulations are indeed coarser than LES, but they resolve gradients at their own grid scale. What is described as "under-resolved" actually corresponds to sub-grid-scale features, which should primarily be handled by the wind farm parameterization (WFP) and the local PBL diffusivity. Ideally, the applied force should account for the local wake expansion and turbine interactions, and then be consistently integrated over the grid cell in a way that accounts for grid-scale wake effects. In the current Fitch WFP, momentum is injected without these considerations. Therefore, the term "under-resolved" seems incorrectly applied to which is a flaw in the WFP.
- 2. In section 4.1, the argument about reduced eddy diffusivity, reduced shear, and less TKE is reasonable, but it would be much stronger if supported by figures. For example, showing eddy diffusivity and shear production (or other MYNN budget terms such as dissipation and advection) would help visualize the interplay you describe. I am not suggesting LES plots comparisons—those simulations are expensive—but mesoscale plots would add clarity and really sustain what you described. You can support the statement of sustained TKE levels via shear production for example.
- 3. Why only Fitch? Including other WFPs, such as Jensen (which claims to represent sub-grid wakes) or Abkar et al. (2015), would provide valuable insights. I am suggesting the latter, because the force applied in the "BASE" experiment is not the same as in the LES even if you are using a  $C_t$  constant in both cases. In Abkar you can control how much force (and consequently TKE) you apply. And if you apply the same force in the mesoscale, would that result in a better recovery? Limiting the study to Fitch sensitivity reduces the scope.
- 4. I commend the authors for using idealized comparisons and matching inflow conditions between mesoscale and LES simulations, including the removal of inertial oscillations.

5. The manuscript does not mention that in WRF, eddy diffusivity  $(K_m)$  is treated separately in the vertical and horizontal directions. Vertical mixing is handled by the PBL scheme, while horizontal mixing is typically represented by a 2D Smagorinsky scheme or constant  $K_m$  under ideal conditions. Since horizontal gradients are discussed, it would strengthen the paper to either show the influence of horizontal  $K_m$  or argue why it is negligible compared to vertical mixing. If  $K_m$  in the vertical is the only important thing then argue around the PBL scheme.

**Specific comments**

- 1. Some statements are vague. For example, line 346: "this resemblance is misleading and fails to represent important physical processes"—which important processes? Be specific. Similarly, line 288: "The bias improves" should be rephrased as "The bias decreases" for clarity.
- 2. Lines 316–320: The paragraph suggests that TKE dissipation inhibits accumulation in LES, yet TKE100 shows more TKE than LES. Please be specific.
- 3. Lines 280–282: What is meant by "momentum entrainment"? Is this a momentum sink or entrainment for wake recovery? Please clarify.
- 4. Treatment of "spikes": These are resolved features. The only valid comparison with mesoscale simulations is the coarsened LES, which looks very good in Fig. 8a,b.
- 5. The same for Fig. 9: Mesoscale results with WFP will naturally be worse than LES at the original resolution, except for inflow conditions, which only support the idea that the WFP is the problem.
- 6. Line 449: You state that slow wake recovery and fast TKE decay effects apply to other NWP and climate models using WFPs. However, in HARMONIE, Fitch performs well (see https://doi.org/10.1029/2021MS002947), possibly due to a different PBL scheme. You can comment in that line of the PBLs.

---

## Author Comment (AC1)

**Response to reviewers' comments**

Paper title: "Under-resolved gradients: slow wake recovery and fast turbulence decay with mesoscale Wind Farm Parameterizations"

under review in *Wind Energy Science*

Regular font – reviewers
**Bold blue font – authors**

**Reviewer #1**

The manuscript "Under-resolved gradients: slow wake recovery and fast turbulence decay with mesoscale Wind Farm Parameterizations" by Radünz et al. compares LES of a regularly spaced idealised offshore wind farm in a conventionally neutral boundary layer with the same wind farm under the same meteorological conditions in an idealised setup of the mesoscale model WRF. In the mesoscale model, different grid resolutions and parameters of the wind farm parametrization are studied. The authors compare these different setups to the LES results and conclude that the wake dissipates slower in the mesoscale model because of under-resolved vertical gradients of wind speed and a decay of TKE that occurs at a shorter distance downstream of the wind farm compared to the LES.

The authors address a relevant topic, as mesoscale modelling has become one of the cornerstones for studying the effect of expanding wind energy deployment on power output and the state of the atmosphere. The manuscript is well written, and the different steps are easy to understand.

**We thank the reviewer for their time and thoughtful review.**

However, I do not think the authors provide enough evidence for their strong conclusion that the mesoscale model is under-representing the wake decay. My doubt lies in a fundamental flaw in the study setup.

Figure 4a clearly shows the difference in power production of the wind turbines in the LES and the mesoscale model. The difference in power production (in the BASE case, about 20% more power in the LES) results in a significant difference in removed momentum between the models. I find it hard to be convinced that the large difference in wake deficit, especially visible in Figures 6a and 8e, is not mainly due to this difference in removed momentum. I would actually claim that Figure 6d reveals that the wake decay is quite comparable except for one grid point just downstream of the wind farm.

**Thank you for the close inspection of the results and for raising this important point. We agree that the higher momentum extraction in the NWP-WFP simulations contributes to the total wind speed deficit within the wind farm, and this effect is discussed in Section 4.2 of the manuscript. However, we emphasize that the slow wake recovery downstream of the wind farm remains a distinct and critical issue, and one of the central focuses of this study.**

**To isolate wake recovery from momentum extraction, we examine wind-speed changes downstream of the wind-farm exit, where the WFP is inactive. Even when wind speeds match at the exit (e.g., TKE100 and aligned LES), the LES recovers substantially faster in the near-farm wake, gaining 0.65–0.8 m/s between 11.2 and 15 km, compared to 0.2–0.5 m/s for the coarser NWP-WFP cases. This leads to a wind-speed bias of 0.15–0.60 m/s attributable solely to slower near-farm wake recovery in the NWP-WFP simulations.**

Because this recovery occurs outside the wind farm, this bias cannot be attributed to momentum extraction or to the specific implementations of the WFPs. Although the divergence develops over a relatively short distance, its impact persists into the far wake and contributes significantly to the overall wind-speed deficit. We therefore argue that slow near-farm wake recovery is as relevant as momentum extraction within the wind farm in shaping mesoscale wake behavior. To clarify this distinction, we have added a paragraph in lines 360–369 of the revised manuscript explicitly quantifying this effect.

"Even though the difference in near-farm wake recovery rates between the NWP-WFP simulations and the LES occurs over a relatively short distance (~1–2 km), it produces measurable bias in wind speeds. Notice that even if some cases match the wind speed (TKE100 and aligned LES) at the farm exit (Fig. 7a), a departure between the two curves occurs downstream. Between the wind farm exit at 11.2 km and 15 km, the gain in wind speed is of about 0.65 m s−1 and 0.8 m s−1 for the aligned and staggered LES (Fig. 7e), respectively. The change in wind speed is much smaller for the coarser resolution NWP-WFP cases, in the range between 0.2 and 0.5 m s−1 (Fig. 7e). On the other hand, the higher resolution cases gain about 0.4 m s−1 (Fig. 8e). Amongst the NWP-WFP simulations, case TKE000 displays the fastest near-farm wake recovery rate because of the largest wind speed deficit. Subtracting the wind speed change of the two LES with the NWP-WFP cases, a difference of 0.15–0.60 m s−1 is found. These values represent the bias in wind speed associated with the slower near-farm wake recovery in the NWP-WFP simulations."

[Figure]

[Figure]

**Figure R1 – New version of Figure 6 that includes the staggered LES reference case and a new metric to evaluate near-farm wake recovery.**

Furthermore, the study is based on a very specific case of wind farm flow. The situation in which all turbines are aligned at close distance to each other and experience full-wake exposure from all upstream turbines is very rare in reality. Much more often, wind turbines operate in partial wake or in the wake of a more distant upstream turbine. In addition, wind farms with regular layouts such as the one studied are rarely found in practice. For this particular case, the mesoscale model stands no chance of replicating the LES power production, as the momentum extracted by each turbine is equally distributed throughout the entire grid-cell cross-section. Thus, I would not recommend basing general conclusions about the mesoscale model's ability to simulate wind-farm flow on this extreme flow case.

We agree with the reviewer that aligned wind farm layouts rarely occur in practice and that the NWP-WFP simulations have a hard time replicating the momentum extraction and power production of the LES. Therefore, we decided to extend our analysis to contemplate the staggered layout case in the LES from the original LES paper (Kasper et al., 2024) (Figure R2). (Which does incorporate the more distant turbine-turbine wake situation raised by the reviewer. This situation is likely more common than that of the aligned layout case, and thus more representative of real wind farm operation.) No changes are necessary to the NWP-WFP simulations at the coarsest resolution (1246 m) because the coarse grid is insensitive to the spanwise displacement of 5$D$/2.

Because the wake losses are mitigated with the staggered layout, the power (Figure R3) and the momentum extraction (Figure R1a) increase compared with the aligned case. Because the wake losses are underestimated in the NWP-WFP simulation due to the averaging of the deficit over a relatively large grid cell, the NWP-WFP simulations tend to agree more with the staggered LES case (as the reviewer had anticipated).

As for the wake recovery, it occurs at a faster rate in the staggered layout. The 'slow wake recovery' of the NWP-WFP simulations that we wish to emphasize is even clearer with the reviewer's suggestion of the staggered layout LES as the reference (Figure R1d). To isolate wake recovery from momentum extraction, we examine wind-speed changes downstream of the wind-farm exit (between 11.2 and 15 km), where the WFP is inactive (Figure R1e). Even when wind speeds match at the exit (e.g., TKE100 and aligned LES), the LES recovers substantially faster in the near-farm wake, gaining 0.65–0.8 m/s between 11.2 and 15 km, compared to 0.2–0.5 m/s for the coarser NWP-WFP cases. The difference in wind speed change between the NWP-WFP simulations with the staggered LES (Figure R1e) has a bias of about 0.39–0.43 m/s. In comparison, the bias is 0.25–0.30 m/s when considering the aligned LES case.

Therefore, this analysis reveals that the mesoscale simulations represent the near-farm wake recovery insufficiently when benchmarked against two distinct wind farm layouts. Our initial conclusion was that the NWP-WFP simulations underestimate wake recovery when benchmarked against the aligned LES case. Now considering a more representative case, we find that the underestimation in wake recovery is even higher. The reviewer's suggestions ultimately helped expand on the conclusions of our work. We are thankful for that.

[Figure]

**Figure R2 – New Figure 6, which includes the staggered LES case as an additional benchmark.**

[Figure]

**Figure R3 – New Figure 4, displaying turbine power, including the staggered LES.**

**Further comments:**

• **R1-1 Title:** I strongly oppose the idea that the wind farm parametrization should be responsible for wake recovery and turbulence calculation downstream of the farm. This should be solely the job of the PBL scheme. Therefore, the title is misleading.

**We agree with the reviewer that it is incorrect that 'the wind farm parametrization should be responsible for wake recovery and turbulence calculation downstream of the farm'. This fact becomes clearer considering how the wake recovery was weakly affected when using a different WFP (MAV). The MAV WFP was included in the revised version of the manuscript to improve our understanding of the role of WFPs in the near-farm recovery. The MAV WFP has improved representation of subgrid wakes, which led to a better prediction of turbine power and momentum extraction compared with the Fitch cases. Despite these improvements, the slow near-farm wake recovery persisted, demonstrating that the issue is not the WFP.**

**Our intention with the title was to inform that a slower wake recovery is occurs in simulations with WFPs because of how the mesoscale grid represents the wake recovery process, and not because the WFPs directly affect wake recovery.**

**But we also see potential for confusion in the title, and so we changed the title to:**

**"Under-resolved gradients: slow wake recovery and low turbulence behind wind farms parameterized in mesoscale simulations"**

**Furthermore, we included more clarifications in the Abstract in lines 14–15:**

**"Furthermore, the slow near-farm wake recovery is not caused by a limitation of the WFP itself, but by the representation of inherently fine-scale processes on a coarse mesoscale grid."**

**and conclusions in lines 591–595:**

**"We note that the slower wake recovery compared with the LES arises from the coarse representation of wake recovery processes at the mesoscale, rather than from deficiencies in the WFPs themselves. This slower wake recovery occurs predominantly in the near-farm wake, a region that lies outside the direct influence of the WFPs and is therefore governed by grid-resolved dynamics and PBL schemes. Naturally, the WFPs exert influence on wake recovery since they affect wind speed and TKE at the wind farm exit."**

• **R1-2 Description of the PBL scheme:** A description of how the PBL scheme uses TKE for calculating mean momentum transport, and how the PBL scheme itself calculates TKE, is missing. This is crucial to understand what is happening in the mesoscale model. In fact, the PBL scheme used is not even mentioned.

**We thank the reviewer for catching this omission. While we had highlighted in the conclusions that future work would consider the use of the 3DPBL scheme, we have now included a description of the MYNN PBL scheme focusing on how it computes TKE and how TKE is used to compute mixing. We also mention the horizontal mixing with the Smagorinsky scheme in lines 176–182.**

**"Vertical turbulent mixing is represented by the MYNN PBL scheme (Nakanishi and Niino, 2009; Olson et al., 2026), while horizontal mixing is treated using a two-dimensional first-order Smagorinsky closure with constant eddy diffusivity (Skamarock et al., 2019). In MYNN, TKE is prognosed from a budget equation that includes shear production, buoyancy production or destruction, vertical turbulent transport, and dissipation (Nakanishi and Niino, 2009; Olson et al., 2026). The diagnosed TKE is then used to compute eddy diffusivities through stability-dependent mixing-length formulations, which directly control the vertical transport of momentum and scalars in the PBL."**

• **R1-3 L. 65 ff.:** I do not like this part of the introduction and Fig. 1. This should be moved entirely to the discussion and conclusion, as it makes assumptions about wake recovery that are actually part of the study. As mentioned earlier, I do not see enough evidence of a consistent underestimation of wake recovery by WRF-WFP. Comparisons with power data (Sanchez-Gomez et al. 2024) show quite good performance.

**We thank the reviewer for this comment and for the opportunity to clarify the intent of this part of the Introduction. Figure 1 is intended to summarize selected results from the literature to provide the reader with a clear depiction of what is commonly described as "slow wake recovery" and "rapid turbulence reduction." We do not make assumptions at this stage; rather, we report general features identified in previous studies (Abkar and Porté-Agel, 2015; Vanderwende et al., 2016; Peña et al., 2022; García-Santiago et al., 2024) to motivate the problem addressed in this work.**

**That said, we agree that there are also investigations reporting good agreement between NWP-WFP simulations and observations, including aircraft transects across wind farms and power data. While these studies typically do not directly evaluate wake recovery rates, they are nevertheless relevant and provide important context.**

We recognize that omitting them from the Introduction resulted in an unbalanced presentation, as they were previously discussed only later in the manuscript. To address this, we have added a paragraph in lines 87–95 that explicitly acknowledges studies showing good agreement with observations, thereby providing a more balanced overview of the current state of the literature.

"However, several studies using NWP-WFP simulations driven by realistic synoptic conditions have reported reasonable agreement between simulated and observed wind speed deficits in wind farm wakes. Such agreement has been demonstrated using aircraft measurements (Siedersleben et al., 2018a, b, 2020; Larsén and Fischereit, 2021; Ali et al., 2023; van Stratum et al., 2022) as well as SCADA data (Sanchez Gomez et al., 2024) in offshore wind farms. The contrast between idealized NWP-WFP simulations evaluated against LES, which can exhibit slower wake recovery, and realistic NWP-WFP simulations validated against observations motivates further investigation. It remains unclear whether the slower wake recovery in idealized configurations arises from limitations of the WFP itself or from deficiencies in the representation of wake recovery on a mesoscale grid. Clarifying this distinction is essential for assessing when NWP-WFP simulations can reliably represent wake evolution and associated power losses, and constitutes the central motivation of this study."

• **R1-4 L. 79:** Montavon et al. (2024) relate the overestimation of power losses mainly to errors in gross-yield calculation, which were addressed by the correction in Vollmer et al. (2024). They find that external and internal wake patterns are well replicated by WRF-WFP. Reported energy-yield differences before this correction should be treated carefully.

We appreciate the review pointing out that the lack of a turbine induction correction is an important source of modeling bias, to which we agree. This becomes clear in the statement in Montavon et al. (2024) about the need for an induction correction: *"WRF on the other hand tends to show power curves which are significantly less energetic (note that the magnitude of this discrepancy appears to be highly sector dependent)."* Our text lacked this information.

We note that Montavon et al. (2024) point to other factors potentially influencing the bias in the simulations: *"We also note that while WRF-WFP (with the Fitch wind farm parametrisation) tends to deliver turbine interaction losses that are consistently larger than those predicted by the engineering models and CFD, we think that there are good reasons to suspect that these losses are overestimated."*

Therefore, we improved our description of their findings on the performance of WRF-WFP in lines 81–85.

Original sentence:

"Ultimately, this slow recovery in NWP-WFP simulations likely contributes to the overestimation of power losses often reported when using WFPs (Lee and Lundquist, 2017; Montavon et al., 2024)."

**Modified sentence:**

**"Ultimately, this slow recovery in NWP-WFP simulations likely contributes to the overestimation of power losses often reported when using WFPs (Lee and Lundquist, 2017; Montavon et al., 2024). In Montavon et al. (2024), additional factors affecting turbine power estimates are identified, including the absence of a turbine induction correction in the standard Fitch WFP, which was subsequently addressed by Vollmer et al. (2024)."**

• **R1-5 L. 231 ff.:** The reason for the different development of power production along turbine rows needs to be explained more precisely, and the consequences for conclusions about wake recovery should be acknowledged.

**That is a great point raised by the reviewer. Upon reviewing our Section 3.2, we noticed that wake recovery and turbine-turbine wake effects are mentioned, but not the momentum extraction by the turbines. This could wrongly lead the reader to think that all patterns in the data are controlled by wake recovery alone. The momentum extraction is key in describing the streamwise evolution of turbine row power. Furthermore, we also included the role of subgrid turbine-turbine wakes in improving power predictions using the MAV WFP:**

**Therefore, we included the following statements in lines 277–282 about the interplay between spatially-averaged wake effects, overestimated power and overestimated momentum extraction and their effect on wake recovery:**

**"In summary, the coarser NWP-WFP configurations, the Fitch WFP spatially averages the wind speed deficit over each grid cell, leading to an underestimation of wake losses at downstream turbines and a corresponding overestimation of power. This excess power implies enhanced momentum extraction within the wind farm, which affects the wind speed at the farm exit. The MAV scheme mitigates this behavior by explicitly accounting for subgrid turbine-turbine wake interactions, resulting in more realistic wake losses and power levels. As a result, the wind speed at the wind farm exit, which is critical for downstream wake recovery, is more accurately represented when subgrid wake effects are included."**

• **R1-6 Figure 5i:** This figure clearly shows one of the weaknesses of the MYNN-PBL scheme at high resolution: there is no horizontal momentum diffusion, so individual turbine wakes persist throughout the domain. The manuscript should focus more on this limitation. I also do not understand why the control volume is smaller in this particular case; it should be the same as in other cases for fair comparison.

**The influence of the PBL scheme on wake recovery is indeed an interesting scope. However, to discuss the vertical diffusion of momentum by the PBL and the horizontal diffusion of momentum by the Smagorinsky scheme in detail would require a strategy focused on parameter sensitivity in both frameworks. Our goal here is to evaluate and**

describe how the wake recovery is represented in idealized mesoscale NWP-WFP simulations. Therefore, we had already highlighted the suggested analysis as future work (iv in the original conclusion, comparison with the 3DPBL scheme).

Regarding the smaller control volume mentioned for the DX2D case, if this refers to the black rectangle shown in the revised Fig. 6e, that rectangle is included solely to indicate the perimeter of the wind farm as represented by the WFP. Owing to the higher horizontal resolution, this perimeter is necessarily smaller. However, the black rectangle is not used for averaging or analysis. For a consistent and fair comparison, all diagnostics for the NWP-WFP simulations and the LES are computed over the same area, defined by the region between the two horizontal dashed lines (revised Figures 5 and 6).

The NWP-WFP simulations at finer resolutions are used to investigate the effect of grid-cell averaging of the wind speed deficit and how these finer wakes influence turbine power. In the paper, we show that cases DX2D and DX2DTKE100 more closely match the LES in terms of power (Figure 4b) and wind speed (Figure 7a).

In lines 234–236 we state the following:

"The high-resolution cases DX2D and DX2DTKE100 are used to assess the impact of sharper wake gradients on power and recovery, acknowledging limitations of the application of traditional PBL schemes at sub-kilometer resolutions due to the terra incognita (Wyngaard, 2004; Rai et al., 2019; Haupt et al., 2019, 2023)."

• R1-7 Figures 6a and 7a: In the LES, the wind recovers by the end of the domain, even exceeding upstream wind speed. In the mesoscale model, the wind never recovers to upstream conditions. Can influences of boundary conditions in the models be excluded?

The LES makes use of periodic boundary conditions with a concurrent precursor method. Correspondingly, the flow is forced back to inflow conditions in the fringe region located at the end of the domain. The fringe region and the small region upstream that is affected by its presence have been omitted from the figures, such that the shown results are not affected by the streamwise boundary. The NWP-WFP simulations also employ periodic boundary conditions, which are then fed to a nested domain with the wind farm. The approaches are shown to produce very similar inflow conditions (Figure 3).

We do indeed observe a minor flow speed-up in the latter portion of the domain. Likely, this is an indirect consequence of the wake deflection, inducing a downward flow of high velocity air to the north side of the wake. Consequently, the wind speed at hub height slightly exceeds the inflow conditions very far downstream. While it is possible that the spanwise boundaries influence this effect, it should be noted that it develops only roughly 20 km downstream of the wind farm, as the wake deflection has to build up first. Importantly, it does not interfere with the near-wake region, which

**is of most interest to our analysis. As for the mesoscale simulations, the wake does not rotate as in the LES, especially the far wake (Figure 6b).**

 • **R1-8 L. 301 ff.:** In the end, there is a pronounced difference in wake recovery exactly one grid cell downwind of the farm (Fig. 6d). It would be interesting to understand this "key mechanism."

**We appreciate the reviewer's interest, and wish to point out that the largest difference in wake recovery occurs in a short distance of a few grid cells, and not at a single grid cell (Figure 4d).**

**The key mechanism for these results is the very large wake recovery rates in the LES near the wind farm exit over a distance of one to a few kilometers only. The larger recovery rates in the LES in the near-farm wake occur because individual turbine wakes are discernible at the farm exit but are quickly mixed out downstream (Figure R2a). The wind speed gradients are largest and the TKE is also very high, which combined create the fast recovery. The difference in recovery between the LES and NWP-WFP simulations over a few grid cells is associated with this mechanism, related to the gradients that are the focus of this manuscript.**

**To better address this point raised by the reviewer, we included another metric to quantify the near-farm wake recovery (Figure R1e). In the LES, the wind speed recovers initially at a faster rate than any of the NWP-WFP simulations with a coarse grid (Figure R1e). Yes, that is a relatively short distance (~1–2 km), but the bias in wind speed in comparison with the aligned and staggered LES cases owing to the slower wake recovery is between 0.15--0.60 m/s. These values are generally smaller than the bias within the wind farm majorly caused by a difference in momentum extraction (0–1 m/s), but are important because their effect is sustained in the far wake. In simple words, the far wake in the NWP-WFP simulations cannot fix the bias in wind speed that was created in the near-farm wake.**

**In Figure R1d,e, the recovery rate for the staggered LES case is even faster, possibly because the momentum extracted is also greater than in the aligned LES case.**

**We included the following paragraph in lines 360–369:**

**"Even though the difference in near-farm wake recovery rates between the NWP-WFP simulations and the LES occurs over a relatively short distance (~1–2 km), it produces measurable bias in wind speeds. Notice that even if some cases match the wind speed (TKE100 and aligned LES) at the farm exit (Fig. 7a), a departure between the two curves occurs downstream. Between the wind farm exit at 11.2 km and 15 km, the gain in wind speed is of about 0.65 m s−1 and 0.8 m s−1 for the aligned and staggered LES (Fig. 7e), respectively. The change in wind speed is much smaller for the coarser resolution NWP-WFP cases, in the range between 0.2 and 0.5 m s−1 (Fig. 7e). On the other hand, the higher resolution cases gain about 0.4 m s−1 (Fig. 8e). Amongst the NWP-WFP simulations, case TKE000 displays the fastest near-farm wake recovery rate because of the largest wind speed deficit. Subtracting the wind speed change of the two LES with the NWP-WFP cases, a difference of 0.15–0.60 m s−1 is found. These**

**values represent the bias in wind speed associated with the slower near-farm wake recovery in the NWP-WFP simulations."**

• **R1-9 Figure 9:** The comparison along one turbine column is misleading. A location exactly between turbine rows would show higher TKE in the mesoscale model than in the LES, as seen in Fig. 8g. A more meaningful comparison would be cross-stream averaged profiles.

**We understand the reviewer's concern regarding the potential for a comparison along a single turbine column to be misleading. Indeed, some panels (former Fig. 9i,j) show locally higher TKE in the LES, which is not representative of spatially averaged or farm-integrated behavior. As shown in former Fig. 6c, several mesoscale cases (e.g., TKE075 and TKE100) exhibit higher TKE than the LES when evaluated using spatially representative averaging.**

**However, unlike former Figs. 6 and 7, which intentionally employ systematic cross-stream and temporal averaging to characterize the bulk wake behavior, Figs. 8 and 9 are designed to do the opposite: they probe local gradients at specific grid cells to assess how NWP-WFP simulations differ structurally from the LES. These local gradients are central to the paper's main argument, namely that the lack of spatial gradients in wind speed in the NWP-WFP compared with the LES slows down the wake recovery in the near-farm region (Equations 1 and 2). To avoid confusion, we have added clarifying text to the manuscript explicitly stating the purpose and limitations of these localized comparisons in lines 400–405:**

**"Figures 9 and 10 intentionally focus on localized profiles at specific turbine-aligned grid cells to highlight differences in resolved gradients between the NWP-WFP simulations and the LES. While spanwise averaging would yield different absolute TKE levels, such averaging would obscure the local shear and turbulence structures that directly control wake recovery in the near-farm region. These figures are therefore not intended to represent farm-averaged behavior, but rather to diagnose the structural deficiencies of mesoscale simulations at the grid-cell scale."**

• **R1-10 L. 510 ff.:** I am not convinced that the difference occurring at exactly one grid cell explains much of the differences in wake deficit elsewhere.

The study contains interesting data with potential to identify key limitations of the PBL scheme in modelling wake recovery comparable to higher-fidelity models. However, I disagree with the manuscript's conclusion that a general slower wake recovery in the mesoscale model is evident from these data. If the manuscript clearly addresses the key limitations of comparing the two datasets and focuses on wake-recovery processes related to the PBL scheme, it could add valuable contributions to the discourse.

**We appreciate this excellent point which made us rethink how we represent how momentum extraction and wake recovery are separated in their influence on the wind speed deficit. Because momentum extraction and wake recovery act simultaneously within the farm, it would be hard to compare them within the farm. Downstream of the**

wind farm, there is no momentum extraction so that the only acting 'agent' is the wake recovery.

In Figure R1e (the new Figure 6), we compute a new metric that measures the change in wind speed having the wind farm exit at 11.2 km as the reference. Shortly after the wind farm exit (the dashed vertical magenta line), the change in wind speed occurs much faster in the aligned LES and especially in the staggered LES. Comparing the gains in wind speed of both the coarsened LES cases with those from the NWP-WFP cases between the wind farm exit at 11.2 km and further downstream at 15 km, we find that the biases in wind speed deficit associated with the slower wake recovery are between 0.25–0.30 m/s (using the aligned LES as reference) and 0.39–0.43 m/s (using the staggered LES as reference). With this approach, we were able to isolate and quantify the slower wake recovery effect from the momentum extraction, which hopefully helps with the reviewer's comment.

Before the end of the near-farm wake region, the bias in wind speed has stabilized and the NWP-WFP simulations display wake recovery rates similar to those of the LES cases. However, the bias associated with the slow wake recovery in the NWP-WFP simulations is generated over a very short distance in the near-farm wake but is important because it is sustained in the far wake.

Regarding the role of the PBL scheme on wake recovery, we improved the description of the MYNN PBL scheme in the vertical transport of momentum. We also mentioned the role of the Smagorinsky scheme in the horizontal transport of momentum in lines 176–182.

"Vertical turbulent mixing is represented by the MYNN PBL scheme (Nakanishi and Niino, 2009; Olson et al., 2026), while horizontal mixing is treated using a two-dimensional first-order Smagorinsky closure with constant eddy diffusivity (Skamarock et al., 2019). In MYNN, TKE is prognosed from a budget equation that includes shear production, buoyancy production or destruction, vertical turbulent transport, and dissipation (Nakanishi and Niino, 2009; Olson et al., 2026). The diagnosed TKE is then used to compute eddy diffusivities through stability-dependent mixing-length formulations, which directly control the vertical transport of momentum and scalars in the PBL."

**Reviewer #2**

This was a difficult review. There are many good contributions in this work: the manuscript is exceptionally well written, the approach is logical, and the figures are really effective. Many simulations had to be conducted, stored, post-processed, and analyzed for this work and I am painfully aware of what that means.

**We thank the reviewer for their time and thoughtful approach to our work, and for the compliments on our presentation and effort.**

But at the end I cannot help but concluding that there is very little new information in it and, actually, quite a few attempts to "sell" new concepts that are really not new and, if anything, misrepresent the real problems.

**Based on the reviewer's comments and suggestions, we have substantially revised the manuscript. We included simulations using an alternative wind farm parameterization, the MAV scheme, which features an improved treatment of subgrid-scale wakes and enables evaluation of wake recovery beyond the Fitch framework. These additions clarify that the slow wake recovery downstream of the wind farm persists even when using an improved WFP and is therefore not attributable to limitations of a specific parameterization. We hope that the revisions below, particularly the inclusion of the MAV framework, help clarify the novelty of our contribution and our emphasis on wake recovery processes rather than wake representation alone.**

1. The title is misleading: by using the plural ("parameterizations") one is led to believe that multiple WFPs are going to be evaluated in the study, whereas only one, that by Fitch et al. (2012), has been assessed. There is no explanation as to why only Fitch's was chosen or why the other three WFPs included in WRF (MAV parameterizations) were not considered. Only the sensitivity to the correction factor a for TKE was evaluated, but that does not change the WFP, still Fitch's. Based on this and the other issues below, I have concluded that this paper is a sensitivity study of the Fitch parameterization to one tuning parameter and cannot be generalized to any other WFP.

**We thank the reviewer for this comment and agree that the first version of the manuscript evaluated only the Fitch wind-farm parameterization with sensitivity explored through the TKE correction factor. We acknowledge that, as written, the plural form in the title could have been interpreted as implying a broader intercomparison than was initially performed.**

**The Fitch scheme was originally selected because it remains the most widely used wind-farm parameterization in WRF-based mesoscale studies and is representative of approaches that impose momentum extraction at the turbine level. We focused on a**

single, commonly used scheme to isolate and diagnose mechanisms governing wake recovery at mesoscale resolution. We acknowledge that using a single WFP, which lacks a more detailed turbine-turbine wake interaction, can limit our scope and understanding of the problem.

As we now show with our inclusion of the MAV scheme, the processes emphasized in our study, especially the evolution of the wake downstream of the wind-farm grid cells where the parameterization is applied, occur outside the direct control of any individual WFP formulation, The behaviors identified here are not unique to the Fitch scheme but arise in other WFPs that impose a momentum deficit without adequately resolving downstream shear production. We agree that this broader applicability was not explicitly demonstrated in the initial study, and we appreciate the reviewers' suggestion to include MAV.

Adopting the reviewer's helpful suggestion to extend the analysis to include other WFPs, we implemented the MAV WFPs (Ma et al., 2022a; Ma et al., 2022b) into WRF v. 4.4 (the version we use here), ported from the standard release of WRF v. 4.6. As explained in the new lines 210-215, one of the MAV WFPs possible setups uses the Xie and Archer (2015) analytical wake model (XA) and a superposition of the hub-height wind speed due for wake overlapping (Ma et al., 2022b). This MAV setup was shown to be superior to Fitch at representing turbine power against SCADA data from an offshore wind farm (Ma et al., 2022b):

"To investigate the role of subgrid wake effects on wake recovery, we additionally implemented the "MAV" wind farm parameterizations (Ma et al., 2022b, a), ported from the standard release of WRF v4.6. The MAV WFP is based on the analytical wake model of Xie and Archer (2015) and represents wake overlap through a superposition of hub-height wind speed deficits (Ma et al., 2022a). In contrast to the Fitch scheme, which relies on grid-cell-averaged momentum extraction, MAV explicitly accounts for subgrid wake interactions and has been shown to improve turbine power predictions when compared against offshore SCADA data (Ma et al., 2022a)."

We simulate two cases with the MAV WFP. The first case (denoted MAV) is simply the BASE case using, instead of Fitch, MAV with the XA wake model and the superposition of hub height wind speed. The second case (MAVDX12D) employs a coarser grid so that more turbines fall within the same grid cells. Fundamentally, the improved representation of the wake effects leads to improved predictions of turbine power and thus momentum extraction. Therefore, the MAV case is superior to the BASE (Fitch) case in representing the wind speed within the farm. Despite the coarser resolution, the results of the MAVDX12D case are similar to those of the MAV case (7D grid spacing), thus demonstrating the consistency of the method.

However, despite these differences in power and momentum extraction within the farm between the BASE and MAV cases, the slower near-farm wake recovery rates persist. Therefore, the issues we report are not exclusive to the Fitch WFP. The results with MAV now appear in revised Figures 4, 6 and 7, and are discussed in Sections 3.2 and 3.4.

[Figure]

[Figure]

**Figure R1 – New version of Figure 6 that includes the staggered LES reference case and a new metric to evaluate near-farm wake recovery.**

[Figure]

**Figure R2 – New version of Figure 2 that includes the staggered LES reference case and MAV WFP simulations. Row-averaged power in the NWP-WFP set of simulations with different wind farm added TKE (a) and grid resolutions (b) compared with the LES.**

**The addition of another WFP expanded the scope of the investigation by demonstrating the different momentum extraction and turbine power computed by multiple WFPs. Furthermore, it reinforced our original point that the slow near-farm wake recovery is only weakly affected by WFP choice and is thus a fundamental feature of the mesoscale grid in NWP simulations.**

**Finally, the title has been accordingly reframed as:**

**"Under-resolved gradients: slow wake recovery and low turbulence behind wind farms parameterized in mesoscale simulations"**

**References:**

**Xie, S., & Archer, C. L. (2015). Self‑similarity and turbulence characteristics of wind turbine wakes via large‑eddy simulation. *Wind Energy*, *18*(10), 1815–1838. https://doi.org/10.1002/we.1792**

**Ma, Y., Archer, C. L., & Vasel-Be-Hagh, A. (2022). The Jensen wind farm parameterization. *Wind Energy Science*, *7*(6), 2407–2431. https://doi.org/10.5194/wes-7-2407-2022**

**Ma, Y., Archer, C. L., & Vasel‑Be‑Hagh, A. (2022). Comparison of individual versus ensemble wind farm parameterizations inclusive of sub‑grid wakes for the WRF model. *Wind Energy*, *25*(9), 1573–1595. https://doi.org/10.1002/we.2758**

2. The paper proposes some sort of a new interpretation of the malfunctioning of the Fitch parameterization: the authors name it "under-resolved gradients". They basically claim that these under-resolved gradients are the fundamental reason why WFPs (but in reality it's just Fitch's) do not resolve the wind speed deficit and added TKE patterns accurately. This is not correct. One could simply replace the term "under-resolved gradients" with "sub-grid wakes"

and none of the proposed findings would be new anymore. It is not the local gradients per se that cause differences in the wake recovery, it is the missing wakes; the local gradients are the obvious and inevitable consequence of having a wake with a localized wind speed deficit. There would be no gradient if there was no wake, obviously. By shifting the attention to the gradients, the true issue disappears, which is that the wake effects need to be parameterized better than how it's done in Fitch's. The wakes are missing, therefore (obviously) the gradients are missing, so let's focus on the missing wakes, not on the missing gradients.

**We appreciate the point raised by the reviewer regarding the under-resolved gradients, WFPs and wake recovery. By using the term under-resolved, we do not claim a new modeling limitation, but rather adopt a convenient description of a well-known concept given typical mesoscale grid resolutions and turbine scales. We examine how this limitation manifests in the context of farm-scale wake recovery in NWP-WFP simulations.**

**When we discuss the under-resolved gradients, we are not pointing out limitations in Fitch, MAV, or any other WFP. We are pointing to a feature that occurs mostly outside the grid cells where the WFP acts, downstream of the wind farm. "Under-resolved gradients" refer to how the wake behind a wind farm is represented in a mesoscale grid and the consequent impacts on wake recovery compared with the LES.**

**We agree with the reviewer that the Fitch WFP in the coarse-resolution cases (BASE, TKE000–100) does not represent the wind speed and TKE levels as well as the LES (Figure 6a,c). This mismatch within the wind farm area is related to the lack of sub-grid wakes, as pointed out by the reviewer. The lack of subgrid wakes leads to higher momentum extraction (Figure 6a) and power (Figure 4a). The improvement in momentum extraction (Figure R1a) and turbine power (Figure R2a) is clear compared with the Fitch-based cases.**

**To better understand the role of WFPs in the momentum extraction and near-farm wake recovery, we expanded our analysis. Following the suggestion by the reviewer, we included simulations using one of the MAV parameterizations with the XA analytical wake model and superposition of the hub-height wind speed. The improvement in momentum extraction and turbine power (Figure R2a) is clear compared with the Fitch-based cases. Despite this improvement, the wake recovery behind the wind farm is still slower in the MAV-based cases than in the LES (Figure R1d,e), a pattern that is shared with the Fitch-based cases.**

**To improve on the points raised by the reviewer, we included statements throughout the manuscript to clarify that the slow wake recovery is not a feature caused by limitations of WFPs (even though WFPs influence the wake). In the Abstract in lines 12–14:**

**"Furthermore, the slow near-farm wake recovery is not caused by a limitation of the WFP itself, but by the representation of inherently fine-resolution processes on a coarse mesoscale grid."**

**In the Conclusions in lines 591–592:**

**"We note that the slower wake recovery compared with the LES arises from the coarse representation of wake recovery processes at the mesoscale, rather than from deficiencies in the WFPs themselves."**

**Finally, we also changed the title to highlight this is not a limitation of the WFPs (the former title had WFPs in it):**

**"Under-resolved gradients: slow wake recovery and low turbulence behind wind farms parameterized in mesoscale simulations"**

**Again, we appreciate the reviewer's pointing to statements that could be potentially misleading to the reader.**

Why are the gradients missing or under-resolved? Because, with the Fitch WFP, the wind speed deficit added by the turbines is smeared in the volume of the grid cell and therefore it is not possible to resolve or create the proper y- or z-gradients. This is an implicit limitation that is obvious and is not the cause, but rather the direct consequence, of the WFP.

**We agree with the reviewer that, in the Fitch WFP, the wind speed deficit is distributed over the grid cell, which necessarily produces weaker horizontal and vertical gradients than those resolved in the LES. We also agree that this behavior is an inherent consequence of how WFPs represent turbine forcing at mesoscale resolution. Our objective is not to present this as a newly identified limitation, but rather to examine how the resulting reduced gradients influence near-farm wake recovery and to clarify the implications of this well-known behavior for wake evolution downstream of the wind farm.**

Furthermore about the gradients: the paper describes them as some sort of an LES feature that is almost undesirable ("the LES … wind speed profiles feature sharp gradients that resemble spikes" or "Once the spikiness disappears, the differences between the [LES and WRF] models stabilizes"). The LES do not simulate just the gradients, they simulate the full wake and therefore gradients appear. The gradients are the effect, not the cause.

**We agree with the reviewer that the gradients observed in the LES are an effect of resolving the turbine wakes, rather than a cause in themselves. Our intent was not to portray these gradients as undesirable or anomalous features of the LES, but to describe the sharp gradients associated with localized wakes that naturally emerge when wake dynamics are fully resolved. We acknowledge that our previous wording, particularly the use of "spikes" or "spikiness," may have conveyed an unintended impression.**

**To address this, we have clarified in the manuscript that these strong localized gradients are an expected consequence of fine-resolution wake-resolving simulations. Specifically, we added the following statement in lines 418–419:**

**"These strong localized gradients are expected with a fine-resolution grid and facilitate faster wake recovery in the LES."**

**In addition, we replaced the terms "spikes" and "spikiness" throughout the manuscript with "strong localized gradients" to improve precision and avoid ambiguity.**

Lastly about the gradients: as described in the manuscript, these gradients are a function of Dy (and Dz), thus a direct comparison between LES and WRF is not correct because the two models have different resolutions and different Dys. But one can look at the magnitude of gradients in the Coarsened LES and compare them to those in WRF. In the figure below, which is a zoomed version of Fig. 5, one can notice that the gradients in WRF on the right (U@1 – U@3)/Dy or (U@2 – U@1)/Dy are actually larger than those in the LES (on the left). The whole discussion in Section 4.1.1 is therefore moot.

[Figure]

**We thank the reviewer for this careful and constructive assessment. We agree that when gradients are evaluated using WRF-scale differencing, locally large instantaneous gradients can occur in the mesoscale model, sometimes exceeding those diagnosed from the coarsened LES. We reiterate that the coarsened LES results are coarsened as a postprocessing step, and are not a LES with a coarse grid. The wake recovery physics still follow the much sharper gradients found in the non-coarsened LES.**

**The intent of Section 4.1.1, however, was not to argue that velocity gradients in WRF are uniformly weaker in magnitude than those in the coarsened LES. Rather, our focus is on the structural, spatial, and temporal organization of wake-induced shear in**

the fully resolved LES compared to its representation in the mesoscale model. While WRF can exhibit strong local gradients over individual grid intervals, these gradients do not organize into a spatially coherent and persistent downstream shear layer comparable to that in the LES, particularly beyond the immediate turbine or farm grid cells.

This distinction is central to our interpretation: shear-driven turbulence production in the far wake depends not only on the instantaneous magnitude of discrete gradients, but on their ability to form sustained, organized shear structures that persist downstream and continuously feed turbulence production. The lack of such coherent structures in the mesoscale representation limits wake recovery, even in the presence of locally strong gradients.

We recognize that this nuance was not sufficiently explicit in the original manuscript. In response, we have revised Section 4.1.1 to clarify that the discussion concerns the organization and persistence of wake-induced shear, rather than the pointwise magnitude of velocity gradients computed at model resolution. The discussion is in lines 492–496:

"While discrete velocity differences evaluated at WRF grid resolution can locally yield gradients comparable to or even exceeding those in a coarsened LES (Fig. 9d), they do not organize into shear structures comparable to those present in the native-resolution LES. In contrast, the LES at native resolution exhibits spatially coherent and persistent shear layers that extend downstream and continuously drive turbulence production and wake recovery, a structural feature that is not maintained in the mesoscale representation."

We thank the reviewer for this comment, which helped sharpen both the interpretation and the presentation of the gradient analysis.

3. The second focus of the paper is the so-called "rapid decay" of TKE in the resolved wake in WRF with Fitch's. This terms is improper because it implicitly assumes that it is sufficient to add TKE in the grid cells of the wind turbines and then the resolved processes will just advect and redistribute this TKE downwind and, as such, the issue is just that this happens too quickly. Basically, the (wrong) assumption here is that the source of TKE is at the turbine. While this is absolutely true for the wind speed deficit and partially true for a single turbine's TKE in the near-wake via tip vortices, it is absolutely not true in the far wake. The authors are in fact aligned with the literature when they recognize the well-known fact that (l. 438-439) "the shear production of TKE depends on spatial gradients, which are under-resolved" but they are incorrect in stating that "TKE fails to persist" there. Instead, it is not FORMED there because of the under-resolved shear. It's not a decay problem, it's a missing addition problem.

The reviewer raises an important and valid distinction. We agree that the terminology "rapid decay of TKE" can be misleading if interpreted as a dissipation-dominated process or as implying that turbine-added TKE should simply persist and advect downstream. We intended to describe the observable downstream behavior of the

**turbine-added TKE in the Fitch scheme, which manifests as a rapid reduction of resolved TKE with distance from the wind farm. The goal was not to describe the physical process per se. However, we agree with the reviewer that this behavior is more accurately interpreted as a lack of sustained TKE production in the near-farm wake. Therefore, we removed the term 'decay' from the manuscript and instead highlighted the insufficient shear production of TKE.**

**As the reviewer notes, and as we state in Section 4.1.2 (lines 510–515), shear production of TKE in the far wake depends on spatial gradients that are under-resolved at the model grid scale. In this context, the apparent "decay" reflects a missing production mechanism associated with unresolved shear, rather than excessive dissipation or advection of turbine-generated TKE. Furthermore, our results confirm that adding more TKE does not completely solve the problem, foreshadowing the underlying causes related with the under-resolved gradients.**

**We revised the manuscript in many places to clarify this point and to avoid language that implies that the problem is one of rapid decay alone. For instance, we replaced "fails to persist" by "is insufficiently generated" in lines 513–515:**

**"Rather, the limitation emerges in the intra- and near-farm wake, where TKE is insufficiently generated because shear production depends on spatial gradients (Mellor and Yamada, 1982; Nakanishi and Niino, 2009) that are under-resolved at mesoscale resolution."**

4. Ultimately the true issue is: what are the authors proposing to resolve these issues? How can we resolve these gradients better? After all these computationally expensive and time consuming simulations to, in my opinion, demonstrate the obvious, what is the solution to the issue? None is proposed. I do not see the ultimate value of this paper to the scientific community, it fails to show novelty except semantically, and it does not provide guidance, let alone a solution, on how to mitigate the issue.

**We appreciate the reviewer's candid assessment and agree that a key question for the community is how these deficiencies in representing shear-driven turbulence in wind-farm wakes can ultimately be addressed.**

**The primary objective of this study is not to present a finalized solution or new parameterization, but rather (focusing on wake recovery) to *diagnose the physical origin of the deficiency* in the wake recovery in NWP simulations that use WFPs and to demonstrate why commonly assumed remedies, like increasing the magnitude of turbine-added TKE, are insufficient when shear gradients remain under-resolved.**

**The wind farm wake is central to these evaluations, and so understanding how NWP-WFP simulations represent wake recovery is crucial. Our scientific goal is not to display the coarsened gradients in mesoscale simulations using WFPs, which is a rather logical outcome, as pointed out by the reviewer. Rather, our investigation revealed what causes the slower near-farm recovery, and the**

coarsened gradients play an important role. This understanding is a unique contribution to the literature and a small piece of the puzzle in addressing the representation of wind farm wakes in NWP-WFP simulations. Since the under-resolved gradients also exist outside the wind farm area where the WFP does not act over the grid cells, our investigation indicates that a solution could involve a subgrid model that acts outside the grid cells with turbines, rather than addressing only the parameterization for the grid cells with turbines.

Thus, we have improved the motivation of our work in the Introduction in a paragraph that brings the contrasting literature where the NWP-WFP reportedly performs well in lines 87–95:

"However, several studies using NWP-WFP simulations driven by realistic synoptic conditions have reported reasonable agreement between simulated and observed wind speed deficits in wind farm wakes. Such agreement has been demonstrated using aircraft measurements (Siedersleben et al., 2018a, b, 2020; Larsén and Fischereit, 2021; Ali et al., 2023; van Stratum et al., 2022) as well as SCADA data (Sanchez Gomez et al., 2024) in offshore wind farms. The contrast between idealized NWP-WFP simulations evaluated against LES, which can exhibit slower wake recovery, and realistic NWP-WFP simulations validated against observations motivates further investigation. It remains unclear whether the slower wake recovery in idealized configurations arises from limitations of the WFP itself or from deficiencies in the representation of wake recovery on a mesoscale grid. Clarifying this distinction is essential for assessing when NWP-WFP simulations can reliably represent wake evolution and associated power losses, and constitutes the central motivation of this study."

In addition, we addressed the smaller TKE in the near-farm wake in comparison with the LES and connected this feature with the lack of spatial gradients. As the reviewer points out, others in the literature acknowledged the impact of lacking gradients on the shear production of TKE. However, here we discuss the subject with a focus on the wake recovery and provide new evidence about the performance of NWP-WFP simulations against LES. We modified the Abstract in lines 2–4 to better convey this research focus:

Original sentence:

"This study evaluates the strengths and limitations of the NWP-WFP approach in the Weather Research and Forecasting (WRF) model.."

Modified sentence:

"This study evaluates the wake recovery behind a wind farm represented by the NWP-WFP approach in the Weather Research and Forecasting (WRF) model.."

As an implication, our results indicate that any effective mitigation strategy must address the representation of wake shear and its interaction with grid-scale dynamics, rather than relying solely on local or turbine-based TKE sources. This requirement has direct implications for the development of

**scale-aware wake parameterizations, hybrid approaches that couple wake models to resolved gradients, or grid-dependent formulations of turbulence production (like the 3D PBL scheme that we suggest in the conclusions).**

**Furthermore, we respectfully submit that by isolating the controlling mechanism, especially in our revised version that incorporates the MAV parameterization to demonstrate that this problem is not unique to Fitch, this study provides value to the community by helping redirect future research and model development efforts toward physically consistent solutions.**

**To clarify this contribution, we have revised the manuscript to more explicitly articulate the guidance emerging from this work. We view this paper as a necessary step toward informed parameterization development rather than an endpoint solution.**

**Furthermore, we further clarified the findings in the Conclusions in lines 591–595:**

**"We note that the slower wake recovery compared with the LES arises from the coarse representation of wake recovery processes at the mesoscale, rather than from deficiencies in the wind farm parameterizations themselves. This slowdown occurs predominantly in the near-farm wake, a region that lies outside the direct influence of the WFPs and is therefore governed by grid-resolved dynamics and PBL schemes. Naturally, the WFPs exert influence on wake recovery since they affect wind speed and TKE at the wind farm exit."**

**We also describe potential solutions in lines 598–600:**

**"A potential path forward is the development of a subgrid model that accounts for the missing spatial gradients driving wake recovery. Such a model would act over grid cells downstream of the wind farm in the near-farm wake and be evaluated against LES, potentially using empirical tuning."**

5. As a final note, I find it extremely inaccurate to state that: "Beyond improving the subgrid parameterizations of momentum sinks and TKE sources, it is equally important to develop new strategies that account for the effects of under-resolved gradients in NWP-WFP frameworks." Again, the gradients are under-resolved precisely because the wakes are not well parameterized. These are not two separate issues, they are the same issue. By focusing the attention on one of the effects of the problem, i.e., the gradients, rather than the root cause, i.e., the misrepresented wakes, this paper ultimately has no use and therefore should not be published.

**We thank the reviewer for raising this important point regarding the interpretation of our concluding statements. Our intent in the concluding discussion was to emphasize where the missing wake physics manifests in mesoscale simulations, namely through insufficiently represented horizontal and vertical shear that ultimately limits near-farm wake recovery. In this sense, the gradients are not an "effect" separate from wake misrepresentation, but**

**rather the physical mechanism through which deficiencies in wake parameterizations impact the resolved flow. We regret that this distinction was not sufficiently clear in the original wording.**

**When referring to strategies that account for under-resolved gradients, we are not advocating an approach that bypasses wake physics. Rather, we emphasize that any successful wake parameterization at mesoscale resolution must ultimately influence turbulence and momentum exchange in a way that reflects wake-induced shear, even when wakes are not explicitly resolved. The representation of these gradients is therefore inseparable from the representation of wakes themselves, particularly in coarse-grid frameworks where wakes transition rapidly from parameterized to resolved scales.**

**Our motivation for this statement arises from the persistent slow wake recovery observed immediately downstream of the wind farm, in grid cells where the WFP is inactive. This behavior cannot be addressed solely by modifying momentum sinks or TKE source terms within the wind farm, as demonstrated by the MAV case, which incorporates improved subgrid wake representation yet exhibits a similar downstream recovery deficit. We therefore believe there is scientific value in documenting a source of modeling bias in the near-farm wake that persists across different WFP formulations and is intrinsically linked to the mesoscale representation of wake recovery.**

**To avoid misinterpretation and better reflect this intent, we have revised the statement in the manuscript as follows:**

**"While improving parameterizations of momentum sinks and TKE sources, it is also important to consider solutions for the region immediately downstream of the wind farm, where WFPs are inactive and wake recovery remains under-represented."**

Minor issue

Figure 5: there might be some errors here. First of all, why is TKE zero in the black column right upstream of the wind farm in panel (h)? For TKE to be zero, added TKE would need to be negative, is this even possible? In panel (l), I would expect a very small error in the last column of the wind farm in the center row where the BASE case shows a grid cell with the highest TKE with the same yellow shade as the Coarsened LES in (d). Why is it in the darkest blue instead, indicating the largest error? The errors should also be smallest along the center row, but there is no such feature

**The average TKE over the grid cells immediately upstream of the wind farm in former Figure 5h is indeed much lower than the ambient levels, being about 0.1 m²/s², but not zero.**

The reviewer provides a fair question. The error colormap in Figure 5l is correct. Indeed, the TKE is largest along the central grid cells of the wind farm in the BASE case, since there are two turbines per grid cell there, thus producing more TKE. Because the grid of the BASE case is used for averaging the LES data (coarsened LES), the central grid cells similarly include two turbines, which produce more TKE. The colormap employed in Figure 5d does not distinguish these differences very well, but looking at the first 2–3 rows, we can see that the TKE is also largest in the central grid cells in the coarsened LES results.

[Figure]

We have modified Figure 5 to account for the issues raised by the reviewer by adjusting the colormap to improve the representation of spatial variations in the results. Now, the larger TKE in the central grid cells in panels (d) and (j) is clear.

[Figure]

**Figure 5 – New Figure 5, including the staggered LES and improved TKE colormap.**

**Reviewer #3**

The paper is generally well written and structured. The figures, results, and comparisons between simulations and reference data are strong. I compliment the authors for the framework and the simulation effort.

**We thank the reviewer for their time and thoughtful review.**

However, the title, the claims, and the arguments in the discussion and conclusions are not sufficiently supported by the presented results.

If the study were reframed to focus on a more specific aspect—such as "Effect of the added TKE on wake recovery in the Fitch parameterization," to give an example—I would immediately recommend acceptance. In its current form, the manuscript does not fully reflect what is stated in the title and abstract. Substantial content changes are needed to align objectives, title, and conclusions.

**Based on the reviewer's comments and suggestions, we have substantially revised the manuscript. We added a new LES of a wind farm with a staggered layout, which provides a more representative reference than the aligned configuration alone. In addition, we included simulations using an alternative wind farm parameterization, the MAV scheme, which features an improved treatment of subgrid-scale wakes and enables evaluation of wake recovery beyond the Fitch framework. We clarified that the slow wake recovery downstream of the wind farm persists when using the MAV WFP and is therefore not attributable to limitations of the WFPs themselves. Finally, we made a concerted effort to ensure consistency across the Title, Abstract, Introduction, and Conclusions, and we supported the discussion of MYNN TKE budget terms with a new Appendix B.**

Below is a list of major and specific comments:

**Major comments**

1. Throughout the paper, mesoscale resolution is presented as the main cause of what is referred to as "under-resolved" gradients. This interpretation is problematic because coarse resolution effects are expected—you cannot resolve something smaller than your grid. Mesoscale simulations are indeed coarser than LES, but they resolve gradients at their own grid scale. What is described as "under-resolved" actually corresponds to sub-grid-scale features, which should primarily be handled by the wind farm parameterization (WFP) and the local PBL diffusivity. Ideally, the applied force should account for local wake expansion and turbine interactions, and then be consistently integrated over the grid cell in a way that accounts for grid-scale wake effects. In the current Fitch WFP, momentum is injected without these considerations. Therefore, the term "under-resolved" seems incorrectly applied to

something that is actually a flaw in the WFP.

We thank the reviewer for this thoughtful comment and for the opportunity to clarify our use of terminology. We agree that mesoscale simulations necessarily resolve gradients at their own grid scale, and that processes smaller than the grid spacing must be treated through parameterizations.

Our use of the term under-resolved is not intended to suggest a new limitation of mesoscale modeling, nor to imply that these gradients should be explicitly resolved at mesoscale resolution. Rather, we use under-resolved to distinguish grid-scale gradients that are present but substantially weaker than those in the LES from truly unresolved subgrid-scale processes, such as intra-grid-cell turbine wakes and wake expansion, which are handled by the WFP. This distinction is introduced explicitly in Section 4.1.

Importantly, our focus is not limited to the wind-farm interior, where the WFP actively represents subgrid-scale forcing, but extends to the near-farm wake region, where the WFP is inactive. In this region, wind speed gradients are considerably weaker in the mesoscale simulations than in the LES, despite being grid-resolved. It is this insufficient representation of gradients, rather than their complete absence, that motivates the term under-resolved in the present context.

We agree with the reviewer that limitations in the Fitch WFP formulation can affect momentum extraction and subgrid-scale wake representation. To address this, we expanded the analysis by implementing the MAV WFP (Ma et al., 2022a,b) in WRF v4.4. The MAV scheme incorporates an analytical wake model and wake superposition and has been shown to improve power predictions relative to SCADA data.

We simulate two cases with the MAV WFP. The first case (denoted MAV) is simply the BASE case using MAV instead of Fitch with the XA wake model and the superposition of hub height wind speed. The second case (MAVDX12D) employs a coarser grid so that more turbines fall within the same grid cells. With these cases, we can evaluate the strengths and weaknesses of the MAV WFP.

While momentum extraction is indeed improved in the MAV cases relative to the BASE (Fitch) case, the near-farm wake recovery remains weakly affected (Figure R1a,d,e). This result indicates that the slow near-farm wake recovery is not a consequence of WFP formulation, but is linked to how wake-induced gradients are represented on a mesoscale grid once the WFP forcing is no longer active.

[Figure]

[Figure]

**Figure R1 – New version of Figure 6 that includes the staggered LES reference case and a new metric to evaluate near-farm wake recovery.**

**References:**

Xie, S., & Archer, C. (2015). Self‑similarity and turbulence characteristics of wind turbine wakes via large‑eddy simulation. *Wind Energy*, *18*(10), 1815–1838. https://doi.org/10.1002/we.1792

Ma, Y., Archer, C. L., & Vasel-Be-Hagh, A. (2022). The Jensen wind farm parameterization. *Wind Energy Science*, *7*(6), 2407–2431. https://doi.org/10.5194/wes-7-2407-2022

Ma, Y., Archer, C. L., & Vasel‑Be‑Hagh, A. (2022). Comparison of individual versus ensemble wind farm parameterizations inclusive of sub‑grid wakes for the WRF model. *Wind Energy*, *25*(9), 1573–1595. https://doi.org/10.1002/we.2758

2. In Section 4.1, the argument about reduced eddy diffusivity, reduced shear, and less TKE is reasonable, but it would be much stronger if supported by figures. For example, showing eddy diffusivity and shear production (or other MYNN budget terms such as dissipation and advection) would help visualize the interplay being described. I am not suggesting LES plots—those simulations are expensive—but mesoscale plots would add clarity and support the statements about sustained TKE levels via shear production.

We appreciate this constructive suggestion and acknowledgement of the additional burden of an LES-base analysis. We agree that mesoscale diagnostics would strengthen the physical interpretation presented in Section 4.1. The intent of that section is to describe the interplay among shear, eddy diffusivity, and TKE within the MYNN framework, but we acknowledge that this discussion is currently supported primarily by inference rather than direct visualization.

In response, we have added an Appendix figure showing the streamwise evolution of selected MYNN TKE budget terms at hub height for the mesoscale simulations (BASE, TKE100, and MAV). The budget terms are computed using the same spanwise averaging applied in Figs. 7 and 8 of the revised manuscript. For clarity, buoyancy production and vertical transport are omitted, as they are small compared to the dominant terms shown here and do not affect the interpretation discussed below.

The Figure R2 shows that TKE peaks within the wind farm due to direct addition by the wind farm parameterizations, with dissipation closely following the TKE magnitude. Shear production increases progressively within the farm as the wind speed deficit develops. Downstream in the near-farm wake, shear production in the BASE and MAV cases exceeds that of the TKE100 case, despite similar TKE and dissipation at the wind-farm exit. This difference is critical: the reduced shear production in TKE100 leads to a faster decay of TKE compared to BASE and MAV. Consistently, cases with larger wind speed deficits exhibit stronger shear production, with the ordering at the farm exit being BASE, MAV, and then TKE100. These results demonstrate the central role of shear production in modulating TKE levels and, consequently, eddy viscosity in the mesoscale wind-farm wake.

We appreciate the reviewer's suggestion, which has helped us identify a clear way to strengthen the presentation without requiring additional simulations.

[Figure]

**Figure R2 – Streamwise variation of spanwise averages over the wind farm area of rotor top tip TKE (a), shear production (b), and dissipation (c) for the BASE, TKE100 and MAV cases. The colored background areas indicate the streamwise extent of the intra-farm wake (gray), near-farm wake (green), and far wake of the farm (blue). The red vertical dashed line marks the wind farm exit.**

3. Why only Fitch? Including other WFPs—such as Jensen (which attempts to represent sub-grid wakes) or Abkar et al. (2015)—would provide valuable insights. I suggest the latter, because the force applied in the "BASE" experiment is not the same as in the LES even though you use a constant Ct in both cases. In Abkar, you can control how much force (and consequently TKE) is applied. If you applied the same force in the mesoscale model, would wake recovery improve? Limiting the study to Fitch sensitivity reduces the scope.

**We appreciate the reviewer's questioning the usage of a single WFP (Fitch) and if the wake recovery would be sensitive to a different forcing by other WFPs.**

**Originally, we included only Fitch because the key scientific question is the representation of the wake recovery process by the mesoscale simulations downstream of the wind farm, and thus outside the influence of the WFP. Also, because that was the only WFP available in the WRF version we used (v. 4.4). The reviewer is right to point at the influence of other WFPs in the forcing within the farm**

and ultimately the wake recovery. As already mentioned in the first comment, we implemented the MAV WFPs (Ma et al., 2022a; Ma et al., 2022b) into our WRF v. 4.4, ported from the standard release of WRF v. 4.6. This approach was necessary because our WRF v. 4.4 is modified for idealized simulations.

The MAV setup uses the Xie and Archer (2015) analytical wake model (XA) to account for subgrid and inter-cell wake effects between individual turbines with a superposition of the hub-height wind speed due to wake overlapping (Ma et al., 2022b). This MAV setup was shown to be superior to Fitch at representing turbine power against SCADA data from an offshore wind farm (Ma et al., 2022b).

We simulate two cases with the MAV WFP. The first case (denoted MAV) is simply the BASE case using MAV instead of Fitch with the XA wake model and the superposition of hub height wind speed. The second case (MAVDX12D) employs a coarser grid so that more turbines fall within the same grid cells. With these cases, we can evaluate the strengths and weaknesses of the MAV WFP.

In Figure 6a,d from the revised manuscript, recently incorporated results using the MAV parameterization are better than Fitch's BASE case in predicting the momentum extraction within the farm, thus agreeing better with the LES. This can be attributed to the better representation of individual turbine wakes in the subgrid wake model (which Fitch lacks).

However, even with an improved treatment of the momentum extraction by the MAV WFP, the slow near-farm wake recovery persists. Answering the question *"If you applied the same force in the mesoscale model, would wake recovery improve?"*, we would have to test this hypothesis to evaluate possible changes in wake recovery. Nonetheless, we have demonstrated that the wake recovery is weakly affected by MAV WFP, which applies an improved forcing compared with the Fitch WFP.

Again, we appreciate the reviewer's suggestion of expanding the usage of WFPs to evaluate their effect on wake recovery.

References:

Xie, S., & Archer, C. (2015). Self-similarity and turbulence characteristics of wind turbine wakes via large-eddy simulation. *Wind Energy, 18*(10), 1815–1838. https://doi.org/10.1002/we.1792

Ma, Y., Archer, C. L., & Vasel-Be-Hagh, A. (2022). The Jensen wind farm parameterization. *Wind Energy Science, 7*(6), 2407–2431. https://doi.org/10.5194/wes-7-2407-2022

Ma, Y., Archer, C. L., & Vasel-Be-Hagh, A. (2022). Comparison of individual versus ensemble wind farm parameterizations inclusive of sub-grid wakes for the WRF model. *Wind Energy, 25*(9), 1573–1595. https://doi.org/10.1002/we.2758

4. I commend the authors for using idealized comparisons and matching inflow conditions between mesoscale and LES simulations, including the removal of inertial oscillations.

**Thank you. A key component in this type of study is matching the inflow conditions.**

5. The manuscript does not mention that in WRF, eddy diffusivity (Km) is treated separately in the vertical and horizontal directions. Vertical mixing is handled by the PBL scheme, while horizontal mixing is typically represented by a 2D Smagorinsky scheme or a constant Km under ideal conditions. Since horizontal gradients are discussed, it would strengthen the paper to either show the influence of horizontal Km or argue why it is negligible compared to vertical mixing. If vertical Km is the only important contributor, then the discussion should clearly focus on the PBL scheme.

**We thank the reviewer for highlighting the distinction between vertical and horizontal turbulent mixing in WRF. In response, we have expanded the model description to explicitly describe how turbulent momentum transport is treated in both directions. Specifically, we clarified that vertical mixing is handled by the MYNN PBL scheme, while horizontal mixing is represented separately using a two-dimensional first-order Smagorinsky closure with constant eddy diffusivity.**

**To address the reviewer's concern regarding wake recovery, we further describe how MYNN predicts TKE and how this TKE is used to compute eddy diffusivities that govern vertical momentum and scalar transport in the PBL. Because near-farm wake recovery in our simulations is primarily controlled by vertical turbulent transport, the discussion focuses on the role of the PBL scheme, while the contribution of horizontal diffusion is expected to be secondary for the cases considered. These clarifications are now included in lines 176–182.**

**"Vertical turbulent mixing is represented by the MYNN PBL scheme (Nakanishi and Niino, 2009; Olson et al., 2026), while horizontal mixing is treated using a two-dimensional first-order Smagorinsky closure with constant eddy diffusivity (Skamarock et al., 2019). In MYNN, TKE is prognosed from a budget equation that includes shear production, buoyancy production or destruction, vertical turbulent transport, and dissipation (Nakanishi and Niino, 2009; Olson et al., 2026). The diagnosed TKE is then used to compute eddy diffusivities through stability-dependent mixing-length formulations, which directly control the vertical transport of momentum and scalars in the PBL."**

**Specific comments**

1. Some statements are vague. For example, line 346: "this resemblance is misleading and fails to represent important physical processes"—which important processes? Be

specific. Similarly, line 288: "The bias improves" should be rephrased as "The bias decreases" for clarity.

**Reevaluating, "physical processes" is not the most appropriate term here. What we meant by physical processes in the original manuscript are the sharp gradients that appear only in the LES at full-resolution. The sharp gradients are explained in the following sentence, but we agree that the physical processes are unclear. Furthermore, we switched the term "spikes" for "strong localized gradients", which are more precise. We modified the sentences in lines 416–418.**

**Original sentence:**

**"While the coarsened LES profile appears similar to those of the NWP-WFP cases within the wind farm ($0 \leq X \leq 10$ km, $0 \leq X/D \leq 56$), this resemblance is misleading and fails to represent important physical processes. The full-resolution LES wind speed profiles feature sharp gradients that resemble spikes (Figs.9a–c)."**

**Modified sentence:**

**"While the coarsened LES profile appears similar to those of the NWP-WFP cases within the wind farm ($0 \leq X \leq 10$ km, $0 \leq X/D \leq 56$), this similarity hides strong localized gradients in the wakes that only appear in the full-resolution LES (Figs.9a–c).".**

**Finally, the statements with "the bias improves" have been corrected.**

2. Lines 316–320: The paragraph suggests that TKE dissipation inhibits accumulation in LES, yet TKE100 shows more TKE than LES. Please clarify.

**Accumulation here refers to a gradual increase in TKE in the streamwise direction within the wind farm. In the NWP-WFP simulations, the TKE rises fast in the first 2 km within the farm and remains nearly constant or gently decreases (cases TKE050–TKE100). Therefore, even though some cases have more TKE within the wind farm than the LES (cases TKE075 and TKE100), the streamwise variation changes. Later on in lines 510–515, this apparent decrease in TKE is associated with the lack of shear production in the NWP-WFP simulations, and not with a dissipation that is too high:**

**"A compelling hypothesis is that the rapid reduction of TKE is not merely a dissipation artifact, but instead a consequence of under-resolved gradients. This interpretation is supported by the close agreement between NWP-WFP and LES TKE levels in the inflow (Fig. 3d) and in the far wake (Fig. 7c), indicating that the mesoscale model does not inherently over-dissipate TKE. Rather, the limitation emerges in the intra- and near-farm wake, where TKE is insufficiently generated because shear production depends on spatial gradients (Mellor and Yamada, 1982; Nakanishi and Niino, 2009) that are under-resolved at mesoscale resolution."**

3. Lines 280–282: What is meant by "momentum entrainment"? Is this a momentum sink, or entrainment contributing to wake recovery? Please clarify.

**The sentence is supposed to mean that momentum entrainment into the wake contributes to wake recovery. Many modifications were made to Section 3.4 during the review, and we modified the original paragraphs that contained such statements. We clarified what we meant with "momentum entrainment" in lines 326–331:**

**"In the intra-farm wake region, the momentum extraction by the turbines acts simultaneously with the wake recovery. The coarse Fitch simulations slightly underestimate the wind speed deficit in the first three turbine rows of the wind farm but overestimate the deficit further downstream (Figs. 7a) relative to the aligned LES. However, using the staggered LES as reference produces much better agreement in the second half of the wind farm, especially for the TKE050–100. Notably, the cases with α values of 75% and 100% show clear improvement at the wind farm exit and in the near-wake region (X < 15 km) due to the enhanced momentum entrainment into the wake associated with the larger TKE."**

4. Treatment of "spikes": These are resolved features. The only valid comparison with mesoscale simulations is the coarsened LES, which looks very good in Fig. 8a,b.

**Our understanding is that two different sets of comparisons are necessary in the paper for different reasons.**

**The comparison with the full-resolution LES describes the local grid-resolved gradients that physically drive wake recovery (Equations 1 and 2). It makes explicit the different representations of the spatial gradients between the LES and NWP-WFP simulations. To understand wake recovery we need to evaluate the local momentum fluxes and spatial gradients near the interface between the wake and the flow outside the wake. As such, if we average the LES (coarsened LES) the resulting gradients are not those that drive the wake recovery in the simulation. The wake recovery in the LES is driven by the sharp or spiky gradients (Figure 8 and 9). Looking from another angle, the wake recovery is underestimated in the NWP-WFP simulations because of the coarser spatial gradients in the wake. However, the spatial gradients of the coarse LES (Figure 8) agree well with those from the NWP-WFP simulations. Thus, the spatial gradients from the coarse LES are not suitable for describing the wake recovery in the LES.**

**The key point of the paper is the slower wake recovery in the NWP-WFP simulations. The driving mechanism for the slower wake recovery is associated with grid-resolved gradients (Equations 1 and 2). Using grid-averaged gradients, such as those represented in the coarse LES, would provide a misleading interpretation of the wake recovery process.**

**However, we agree with the reviewer that the comparison against the coarsened LES can be useful, but for the specific purpose of evaluating the mean properties of the wind farm wake, such as wind speed and TKE.**

5. The same applies to Fig. 9: Mesoscale results with WFP will naturally be worse than LES at the original resolution, except for inflow conditions. This only reinforces the idea that the WFP itself is the problem.

**We agree with the reviewer that the NWP-WFP simulations' results are worse than the LES at the original resolution. Showing the reduced spatial gradients in the horizontal (Figure 8) and vertical (Figure 9) directions in the NWP-WFP simulations is essential to explain the slow near-farm wake recovery described by Equations 1 and 2.**

**Even though the Fitch WFP has limitations, the central problem is how the mesoscale grid represents the wake and its recovery (under-resolving the gradients). In Figure 7a,d from the revised manuscript, recently incorporated results using the MAV parameterization are better than Fitch in predicting the momentum extraction within the farm, thus agreeing better with the staggered LES. This can be attributed to the better representation of individual turbine wakes in the subgrid wake model (which Fitch lacks). Nonetheless, the near-farm wake represented in simulations with MAV similarly underestimates the wake recovery rate.**

6. Line 449: You state that slow wake recovery and fast TKE decay effects apply to other NWP and climate models using WFPs. However, in HARMONIE, Fitch performs well (see https://doi.org/10.1029/2021MS002947), possibly due to differences in the PBL scheme. You may comment along those lines regarding the role of PBLs.

**We appreciate the reviewer's comment regarding the role of the PBL scheme in modulating wake recovery. We agree that the PBL scheme is potentially a source of variability on the wake recovery among many studies, that includes Van Stratum et al. (2022). Different PBL schemes compute the mixing coefficients differently, which influences the vertical transport of momentum. Different PBL schemes can also produce different inflow wind profiles, such that both the background wind and the mixing can be different. Other sources of variability are the input data, grid resolution, nesting approach, among other factors.**

**We have included the reference to the HARMONIE project in the Introduction in a paragraph that mentions the literature where the NWP-WFP reportedly performs well in lines 87–95:**

**"However, several studies using NWP-WFP simulations driven by realistic synoptic conditions have reported reasonable agreement between simulated and observed wind speed deficits in wind farm wakes. Such agreement has been demonstrated using aircraft measurements (Siedersleben et al., 2018a, b, 2020; Larsén and Fischereit, 2021; Ali et al., 2023; van Stratum et al., 2022) as well as SCADA data (Sanchez Gomez et al., 2024) in offshore wind farms. The contrast between idealized NWP-WFP simulations evaluated against LES, which can exhibit slower wake recovery, and realistic NWP-WFP simulations validated against observations motivates further investigation. It remains unclear whether the slower wake recovery in idealized configurations arises from limitations of the WFP itself or from deficiencies in the representation of wake recovery on a mesoscale grid. Clarifying**

this distinction is essential for assessing when NWP-WFP simulations can reliably represent wake evolution and associated power losses, and constitutes the central motivation of this study."

We also included a statement about the role of the PBL scheme on the vertical mixing in lines 176–182:

"Vertical turbulent mixing is represented by the MYNN PBL scheme (Nakanishi and Niino, 2009; Olson et al., 2026), while horizontal mixing is treated using a two-dimensional first-order Smagorinsky closure with constant eddy diffusivity (Skamarock et al., 2019). In MYNN, TKE is prognosed from a budget equation that includes shear production, buoyancy production or destruction, vertical turbulent transport, and dissipation (Nakanishi and Niino, 2009; Olson et al., 2026). The diagnosed TKE is then used to compute eddy diffusivities through stability-dependent mixing-length formulations, which directly control the vertical transport of momentum and scalars in the PBL."